# PCGF6-PRC1 suppresses premature differentiation of mouse embryonic stem cells by regulating germ cell-related genes

Mitsuhiro Endoh[1,2,3,4,5]*, Takaho A Endo[6], Jun Shinga[7], Katsuhiko Hayashi[8], Anca Farcas[9], Kit-Wan Ma[1], Shinsuke Ito[1,2], Jafar Sharif[1,2], Tamie Endoh[1,3,5], Naoko Onaga[1], Manabu Nakayama[10], Tomoyuki Ishikura[1], Osamu Masui[1], Benedikt M Kessler[11], Toshio Suda[3,4], Osamu Ohara[6,10], Akihiko Okuda[12], Robert Klose[9], Haruhiko Koseki[1,2]*

[1]Laboratory for Developmental Genetics, RIKEN Center for Integrative Medical Sciences, Yokohama, Japan; [2]Core Research for Evolutional Science and Technology, Yokohama, Japan; [3]Centre for Translational Medicine,Cancer Science Institute of Singapore, National University of Singapore, Singapore, Singapore; [4]International Research Center for Medical Sciences, Kumamoto University, Kumamoto, Japan; [5]Research Institute for Radiation Biology and Medicine, Hiroshima University, Hiroshima, Japan; [6]Laboratory for Integrative Genomics, RIKEN Center for Integrative Medical Sciences, Yokohama, Japan; [7]Laboratory for Immunotherapy, RIKEN Center for Integrative Medical Sciences, Yokohama, Japan; [8]Department of Developmental Stem Cell Biology, Faculty of Medical Sciences, Kyushu University, Fukuoka, Japan; [9]Department of Biochemistry, Oxford University, Oxford, United Kingdom; [10]Chromosome Engineering Team, Department of Technology Development, Kazusa DNA Research Institute, Kisarazu, Japan; [11]Mass Spectrometry Laboratory, Target Discovery Institute, Nuffield Department of Medicine, University of Oxford, Oxford, United Kingdom; [12]Division of Developmental Biology, Research Center for Genomic Medicine, Saitama Medical University, Saitama, Japan

*For correspondence: csime@nus. edu.sg (ME); haruhiko.koseki@ riken.jp (HK)

**Competing interests:** The authors declare that no competing interests exist.

**Abstract** The ring finger protein PCGF6 (polycomb group ring finger 6) interacts with RING1A/B and E2F6 associated factors to form a non-canonical PRC1 (polycomb repressive complex 1) known as PCGF6-PRC1. Here, we demonstrate that PCGF6-PRC1 plays a role in repressing a subset of PRC1 target genes by recruiting RING1B and mediating downstream mono-ubiquitination of histone H2A. PCGF6-PRC1 bound loci are highly enriched for promoters of germ cell-related genes in mouse embryonic stem cells (ESCs). Conditional ablation of *Pcgf6* in ESCs leads to robust de-repression of such germ cell-related genes, in turn affecting cell growth and viability. We also find a role for PCGF6 in pre- and peri-implantation mouse embryonic development. We further show that a heterodimer of the transcription factors MAX and MGA recruits PCGF6 to target loci. PCGF6 thus links sequence specific target recognition by the MAX/MGA complex to PRC1-dependent transcriptional silencing of germ cell-specific genes in pluripotent stem cells.

## Introduction

Polycomb group (PcG) proteins are evolutionarily conserved epigenetic repressors of developmental genes. PcG-mediated gene silencing involves at least two distinct enzymatic activities directed to histone tails: the first mediates Histone H2A mono-ubiquitination at K119 (H2AK119ub1) by the poly-comb repressive complexes 1 (PRC1), while the second mediates H3 tri-methylation at K27 (H3K27me3) by the polycomb repressive complex 2 (PRC2) (*Cao et al., 2002*; *Czermin et al., 2002*; *Kuzmichev et al., 2002*; *Müller et al., 2002*; *Shao et al., 1999*; *Wang et al., 2004*). According to the canonical view, H3K27me3 deposition by the EZH1/2 (enhancer of zeste homologs 1 and 2) his-tone methyltransferases leads to binding of the H3K27me3-reader protein CBX2 (chromobox protein homolog 2), in turn facilitating sequential recruitment of canonical PRC1 (cPRC1) (*Bernstein et al., 2006*; *Cao et al., 2002*; *Czermin et al., 2002*; *Fischle et al., 2003*; *Kuzmichev et al., 2002*). Con-versely, H2AK119ub1 deposition by the non-canonical PRC1 (ncPRC1) promotes downstream recruitment of PRC2 and H3K27me3 (*Blackledge et al., 2014*; *Cooper et al., 2014*). Myriad other accessory molecules interact with both PRC1 and PRC2, conferring robustness and reversibility to PcG-mediated gene repression (*Li et al., 2011*).

The molecular complexity underlying PcG-mediated gene silencing could be partly explained by the diversity of the PCGF factors (from PCGF1 to PCGF6) that directly associate with RING1A/B pro-teins. For example, canonical PRC1 (cPRC1) may include PCGF2 (also known as MEL18: melanoma nuclear protein 18) or PCGF4 (also known as Bmi1: B cell-specific Moloney murine leukemia virus integration site 1). These cPRC1 complexes can be further classified into specific sub-complexes according to their association with CBX (CBX2/4/6/7/8) or PHC (PHC1/2/3: polyhomeotic homologs 1, 2 and 3) (*Gao et al., 2012*; *Vandamme et al., 2011*). CBX proteins contribute to recognition of H3K27me3, and therefore mediate recruitment of cPRC1 into target loci; while PHC proteins medi-ate gene compaction through polymerization of the SAM (sterile alpha motif) domain to facilitate transcriptional silencing (*Isono et al., 2013*).

Non-canonical PRC1 (ncPRC1) associates with PCGF1, PCGF3, PCGF5 or PCGF6 and RYBP (RING1 and YY1-binding protein) or YAF2 (YY1 Associated Factor 2), but does not interact with CBX2/4/6/7/8 or PHC1/2/3 (*Gao et al., 2012*). In addition to the core subunits RING1A/B and PCGF1, ncPRC1-containing PCGF1 (PCGF1-PRC1) also incorporates KDM2B [Lysine (K)-Specific Demethylase 2B], BCOR (BCL6 Corepressor), BCORL1 (BCL6 Corepressor-Like 1), RYBP, YAF2 and SKP1 (S-Phase Kinase-Associated Protein 1) (*Farcas et al., 2012*; *Gearhart et al., 2006*). Non-canon-ical PRC1 complexes are functionally linked with their canonical counterparts (cPRC1), and most of their target genes overlap with each other (*Blackledge et al., 2015*). These overlapping genes often possess unmethylated CpG islands (CGI) in their promoters, and these are preferentially bound by the CXXC (two cysteines separated by two other residues) domain of KDM2B via recognition of unmethylated CpG dinucleotides (*Farcas et al., 2012*; *He et al., 2013*; *Wu et al., 2013*). KDM2B binding to target genes allows direct recruitment of PCGF1-PRC1, followed by binding of PRC2 through recognition of H2AK119ub1 (*Blackledge et al., 2014*; *Cooper et al., 2014*). As PRC2 fur-ther recruits cPRC1, non-canonical PCGF1-PRC1 can therefore activate the PRC2-cPRC1 axis to ensure robust transcriptional silencing of developmental genes that harbor unmethylated CGIs.

Another PCGF homolog, PCGF6, was first identified in a multimeric protein complex associated with the E2F6 transcription factor (*Ogawa et al., 2002*). This complex was annotated as PCGF6-PRC1, and was shown to form stable complexes with several other well-known epigenetic factors such as RING1A/B, PCGF6, L3MBTL2 [lethal(3)malignant brain tumor-like 2], RYBP, YAF2, G9A (also known as EHMT2: euchromatic histone-lysine N-methyltransferase 2), GLP (G9a-like protein 1, also known as EHMT1), and CBX1/3 (*Gao et al., 2012*; *Hauri et al., 2016*; *Kloet et al., 2016*; *Ogawa et al., 2002*; *Qin et al., 2012*; *Trojer et al., 2011*). Interestingly, PCGF6-PRC1 also interacts with sequence-specific DNA binding proteins such as E2F6, MAX, MGA and TFDP1 (transcription factor Dp-1) (*Gao et al., 2012*; *Hauri et al., 2016*; *Kloet et al., 2016*; *Ogawa et al., 2002*; *Qin et al., 2012*; *Trojer et al., 2011*). This suggests that such DNA binding proteins could play a role in sequence specific recruitment of PCGF6-PRC1 to target loci; however, this notion has not been experimentally validated.

In this study, we therefore purified the PCGF6-PRC1 complex and examined the contribution of PCGF6 to ESC maintenance and embryonic development. We demonstrate that PCGF6 mediates repression of target genes by recruiting RING1B and facilitating H2AK119ub1. Taking advantage of

a *Pcgf6* conditional allele, we show that PCGF6 and RING1B common targets are enriched for meiosis- and germ cell-related genes in ESCs, and that such genes are robustly de-repressed in the absence of PCGF6 (*Pcgf6*-KO). Importantly, silencing of germ cell-related genes by PCGF6 likely plays a role in proliferation and growth of ESCs. We further demonstrate that PCGF6 is involved in pre- and peri-implantation mouse development. Indeed, loss of *Pcgf6* leads to pleiotropic defects in vivo, including aberrant axial development and impaired placenta formation. We also reveal a unique recruitment mechanism amongst PRC1 complexes whereby PCGF6-PRC1 utilizes its MGA and MAX components as a heterodimeric DNA binding module to directly recognize and repress expression of germ cell- and meiosis-related genes to support ESC maintenance and embryonic development.

## Results

### PCGF6 forms complexes with PRC1 components

Previous proteomic approaches have repeatedly identified PCGF6 as a component of multimeric protein complexes designated as PCGF6-PRC1 that included MAX, MGA, E2F6, TFDP1, RING1B, RING1A, CBX3, RYBP, L3MBTL2, YAF2 and WDR5 in human cell lines (*Gao et al., 2012*; *Hauri et al., 2016*; *Ogawa et al., 2002*; *Trojer et al., 2011*). To address the composition of PCGF6 complexes in mouse ESCs, we stably expressed an epitope-tagged form of PCGF6 in mouse ESCs and affinity purified it from nuclear extracts, then used LC-MS/MS analysis to identify associated proteins. We observed strong association of PCGF6 with MGA, RING1B, RING1A, CBX3, CBX1, RYBP, L3MBTL2, YAF2 and TFDP1 (*Figure 1A,B*), indicating that the mouse ESC PCGF6 complex is similar to those purified from human cells (*Gao et al., 2012*; *Hauri et al., 2016*; *Kloet et al., 2016*; *Ogawa et al., 2002*; *Trojer et al., 2011*). We however did not detect considerable amounts of MAX in the PCGF6 complexes in mouse ESCs.

   We went on to confirm these results by immunoprecipitation followed by immunoblotting (IP-IB). For this purpose, we stably expressed FLAG-tagged PCGF6 in *Pcgf6*-deficient (*Pcgf6*-KO) ESCs (*Figure 1—figure supplement 1A,B*) or FLAG-RING1B in wildtype (WT) ESCs (*Figure 1C*). We observed exogenous FLAG-PCGF6 expressed at the similar level of endogenous PCGF6. We then tested the interaction of these tagged proteins with endogenous RING1B, RYBP, L3MBTL2 and MAX (*Figure 1C*). We indeed found considerable association of MAX with both FLAG-PCGF6 and FLAG-RING1B as well as the other three proteins. We further observed PCGF1 or PCGF2 was not co-IPed with FLAG-PCGF6 while both of them associated with FLAG-RING1B. We finally confirmed EZH2 or SET1 were not co-IPed with either FLAG-PCGF6 or FLAG-RING1B. These results suggested that PCGF6 would be primarily involved in non-canonical PRC1 (ncPRC1), particularly PCGF6-PRC1, in mouse ESCs.

### PCGF6 shares target genes with RING1B

The catalytic activity of both canonical PRC1 (cPRC1) and ncPRC1 is mediated by RING1A and RING1B (*Buchwald et al., 2006*; *Gearhart et al., 2006*; *Trojer et al., 2011*; *Wang et al., 2004*). Consistent with this model, it has been shown that genetic ablation of *Ring1A/B* leads to widespread disruption of PRC1-dependent gene repression (*de Napoles et al., 2004*; *Endoh et al., 2008*; *Leeb and Wutz, 2007*; *Wang et al., 2004*). To determine whether PCGF6 plays a role in gene repression in association with RING1B, we examined the overlap between PCGF6- and RING1B-bound sites by PCGF6 ChIP-seq (chromatin immunoprecipitation followed by deep sequencing), by using a FLAG-PCGF6 expressing *Pcgf6*-KO ESCs. Our results considerably overlapped with those reported recently (*Figure 1—figure supplement 1C*) (*Yang et al., 2016*). This analysis revealed that PCGF6 bound to the transcriptional start sites (TSSs) of a group of genes associated with germ cell specific functions, such as *Tex19.1* (testis-expressed protein 19A) and *Tdrkh* (Tudor and KH domain containing) (*Figure 1D*). Indeed, these genes were also marked by RING1B, indicating that PCGF6 and RING1B share common targets.

   Intriguingly, cPRC1 targets, such as the *Hoxd* (homeobox protein Hox-D) gene cluster, were not bound by PCGF6 (*Figure 1D*), indicating that PCGF6 might associate with a subset of RING1B-bound genes that are not marked by cPRC1. A detailed examination of ChIP-seq datasets revealed that 1218, 2959 and 4946 genes in ESCs were bound by PCGF6, RING1B, and H3K27me3, respectively (*Figure 1E*). As expected, PCGF6-bound genes partially but significantly overlapped with both

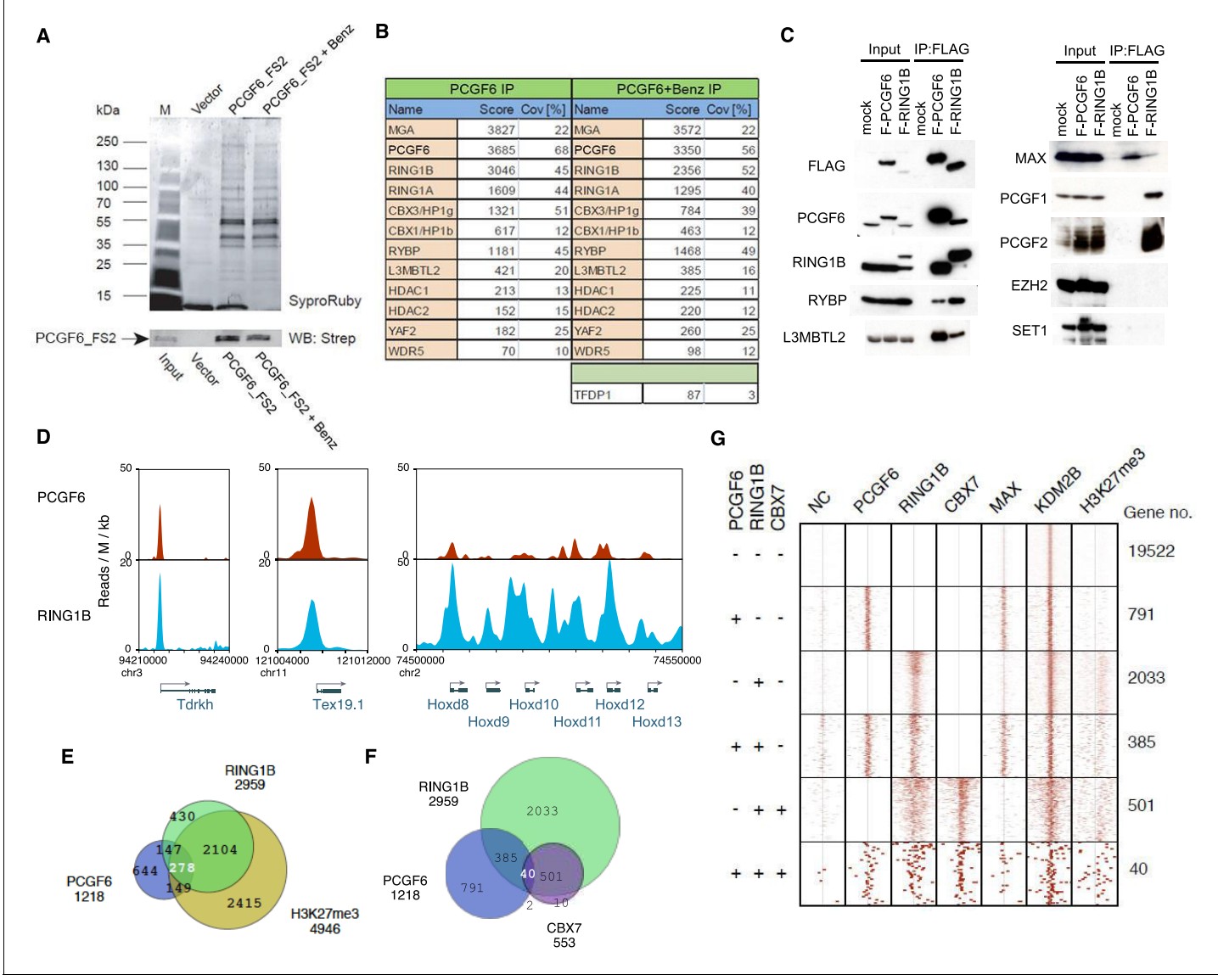

**Figure 1.** Biochemical properties of PCGF6-PRC1 and its target genes in ESCs. (**A**) Affinity purification of PCGF6-containing complexes in ESCs. To purify PCGF6 and associated proteins, a mouse ESC cell line stably expressing Flag-2xStrepII (FS2)-tagged PCGF6 was generated. Nuclear extract was isolated from this cell-line, PCGF6 was affinity purified, and the purified proteins were subjected to mass spectrometry. Purified PCGF6 fractions were resolved by gradient SDS-PAGE and visualized by SyproRuby staining. The purifications were performed in the absence and presence of benzonase (Benz) to exclude DNA-mediated interactions and a cell line containing only the empty vector was used as control for non-specific binding to the affinity matrix. The elutates were probed by western blot for streptavidin (Strep). (**B**) Identification of proteins that form stable complexes with PCGF6 in ESCs. Elutions from the PCGF6 affinity purification were directly analyzed by tryptic digestion followed by peptide identification by LC-MS/MS. The Mascot scores and peptide coverage are shown for the respective affinity purifications. (**C**) Confirmation of PCGF6-containing complexes by immunoprecipitation-immunoblot (IP-IB) analysis. Whole-cell extracts (WCE) from ESCs expressing FLAG-tagged PCGF6 or RINGB were subjected to IP using anti-FLAG antibody. The WCE and immunoprecipitates were separated on SDS-PAGE and analyzed by IB with the indicated antibodies. (**D**) Screenshot views for the distribution of PCGF6 (red) and RING1B (blue) at target genes in ESCs determined by ChIP-seq. The chromosomal positions are indicated on the x-axis. The transcription start sites (TSSs) are denoted by arrows. (**E**) Venn diagram representation for the overlap of PCGF6, RING1B and H3K27me3 target genes in ESCs identified by ChIP-seq. The number of genes bound by PCGF6, RING1B and H3K27me3 and included in each fraction are indicated. (**F**) Venn diagram representing the overlap of PCGF6, RING1B and CBX7 target genes. Published ChIP-seq data for CBX7 was obtained from NCBI GEO (accession number GSM1041373). (**G**) A heat map view for distribution of PCGF6, RING1B, CBX7, MAX, KDM2B and H3K27me3 in ±4 kb genomic regions around transcription start sites (TSS). Genes are classified based on their occupancy by PCGF6, RING1B and CBX7 in ESCs. The signal from a negative control (NC: FLAG-ChIP in mock transfected ESCs) was also shown.

The following source data and figure supplement are available for figure 1:

*Figure 1 continued on next page*

*Figure 1 continued*

**Source data 1.** Raw data for LC-MS/MS analysis shown in *Figure 1B*.
**Figure supplement 1.** Generation of a *Pcgf6* conditional allele and properties of CpG islands at PCGF6-PRC1 target genes.

RING1B and H3K27me3 target genes (*Figure 1E,F*). These observations were further validated by ChIP-qPCR analysis (*Figure 1—figure supplement 1D*).

## PCGF6 binds in close proximity to the TSS of genes and prefers short-CGIs

To clarify whether PCGF6 marked loci represented cPRC1 or ncPRC1 target genes, we compared the list of PCGF6- and CBX7 (Chromobox Homolog 7)-bound genes. As CBX7 is a key component of cPRC1 (*Bernstein et al., 2006*; *Morey et al., 2013*), exclusion of PCGF6-bound sites from CBX7-bound sites would indicate an ncPRC1-centric role for PCGF6, while an overlap would indicate the opposite. CBX7-bound genes were mostly co-occupied by RING1B, as expected (*Figure 1F*). PCGF6 target sites, however, exhibited only a limited overlap with CBX7, supporting the model in which PCGF6 is predominantly involved in ncPRC1.

To gain insight into the binding profile of PCGF6 at target genes, we plotted PCGF6, RING1B, CBX7, MAX, KDM2B and H3K27me3 ChIP-seq reads in a ±4 kb region surrounding TSS (*Figure 1G*). This analysis revealed that PCGF6 bound in close proximity to TSS, unlike RING1B, CBX7 and H3K27me3, all of which exhibited a broader distribution (*Figure 1G*). We observed considerable association of MAX at PCGF6 targets irrespective of RING1B binding. In contrast, H3K27me3 deposition at PCGF6 targets was mainly seen at RING1B-bound genes. We also noted that PCGF6 target genes harbor CGIs that tend to be slightly shorter and more methylated than CBX7 targets (*Figure 1—figure supplement 1E*). Consistently, PCGF6 targets were bound by KDM2B. KDM2B loss induced a subtle increase of RING1B binding at PCGF6 targets while RING1B binding at CBX7 targets was marginally decreased (*Figure 1—figure supplement 1F*). Moreover, PCGF6-bound genes were up-regulated to a much lesser extent than CBX7-bound genes in *Kdm2b*-KO ESCs (*Figure 1—figure supplement 1G*). These results suggest that the CGI recognition by KDM2B has distinct roles for PCGF6-PRC1 recruitment from that reported for cPRC1 (*Farcas et al., 2012*).

## PCGF6 represses RING1B-bound genes

We next wondered whether PCGF6 binding was functionally required for transcriptional repression. To test this, we performed RNA-seq and found that more than 250 genes were considerably up-regulated in *Pcgf6*-KO ESCs (*Figure 2—figure supplement 1A*). Importantly, these genes were a subset of the genes up-regulated in the *Ring1A* and *Ring1B* double knockout (*Ring1A/B*-dKO) ESCs (*Figure 2—figure supplement 1A*). To gain insight into the association of PCGF6 and RING1B for gene regulation, we divided RING1B- or PCGF6-bound genes into three categories (PCGF6+RING1B+, PCGF6-RING1B+ and PCGF6+RING1B-) and examined their expression in WT ESCs using FPKM (fragments per kilobase of exon per million mapped sequence reads) values (*Figure 2—figure supplement 1B*). This analysis showed that the PCGF6+RING1B+ genes were barely transcribed in WT ESCs; likewise, the PCGF6-RING1B+ genes were marginally expressed. In contrast, the PCGF6+RING1B- group that harbors CGIs was robustly expressed. We went on to examine changes of respective gene expression levels in each category in the absence of *Pcgf6* (*Pcgf6*-KO) (*Figure 2A*, left). We found both PCGF6+RING1B+ and PCGF6-RING1B+ genes were significantly up-regulated while expression levels of PCGF6+RING1B- or PCGF6-RING1B- genes were barely altered in *Pcgf6*-KO ESCs (*Figure 2A*, left). We however also found that expression levels of highly up-regulated genes (more than 5-times; indicated by red lines in *Figure 2A*) in the PCGF6+RING1B+ category were considerably higher than those in the PCGF6-RING1B+ category in *Pcgf6*-KO ESCs. Those categories were also up-regulated in *Ring1A/B*-dKO ESCs (*Figure 2A*, right), suggesting a functional link between PCGF6 and RING1A/B for down-regulation of such genes. We further confirmed these changes in expression levels of selected PCGF6+RING1B+ and PCGF6-RING1B+ genes by RT-qPCR in *Pcgf6*-KO and *Ring1A/B*-dKO ESCs (*Figure 2—figure supplement 1B,C*). Together, PCGF6 is

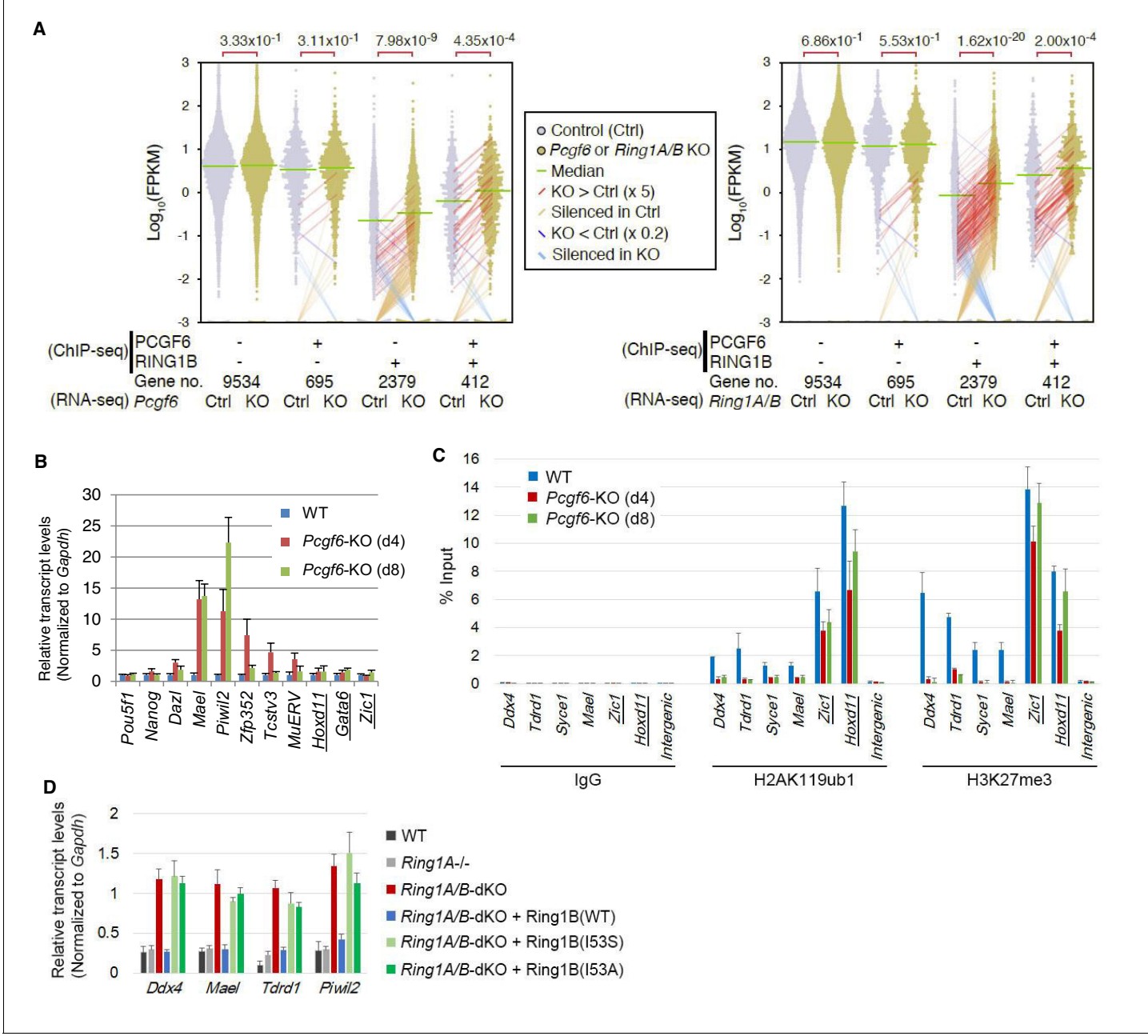

**Figure 2.** RING1B-dependent repression of target genes by PCGF6-PRC1. (A) Dot plot representation for gene expression changes in *Pcgf6*-KO (left) and *Ring1A/B*-dKO (right) ESCs. Expression levels of respective genes (FPKM: determined by RNA-seq) in each group (classified by ChIP-seq) in control (Ctrl) and respective knockouts (KO) are shown by grey and yellow dots, respectively. The median of FPKM in each group is indicated by a green horizontal line. Genes whose expression (FPKM) changed more than 5-fold were indicated by red (up-regulated in KO) or blue (down-regulated in KO) lines. Genes silenced in Ctrl but active in KO are indicated by orange lines, while genes vice versa are indicated by light blue lines. The number of genes included in each subset is shown at the bottom. *p*-values for the difference of expression changes between Ctrl and KO were calculated by the Student's *t*-test and are indicated above each graph. (B) RT-qPCR-based validation for expression changes of selected genes in *Pcgf6*-KO ESCs. Expression levels of genes required for pluripotency, or bound by PCGF6-PRC1 or canonical PRC1 (underlined) in *Pcgf6*$^{fl/fl}$;*Rosa26::CreERT2*$^{tg/+}$ ESCs before (WT) or after OHT treatment (day 4 and 8). Expression levels were normalized to a *Gapdh* control and are depicted as fold change relative to OHT-untreated (WT) ESCs. Error bars represent standard deviation determined from at least three independent experiments. (C) Changes in local H2AK119ub1 and H3K27me3 deposition at selected PCGF6-PRC1 target genes in *Pcgf6*-KO ESCs. Local levels of H2AK119ub1 and H3K27me3 at promoter regions of genes bound by PCGF6-PRC1 or canonical PRC1 (underlined) in *Pcgf6*$^{fl/fl}$;*Rosa26::CreERT2*$^{tg/+}$ ESCs before (WT) or after (day 4 and 8) OHT treatment were determined by ChIP-qPCR analysis. The relative amount of ChIPed DNA is depicted as a percentage of input DNA. Error bars represent standard deviation determined from at least three independent experiments. (D) Requirement of RING1B catalytic activity for repression of

*Figure 2 continued on next page*

*Figure 2 continued*

genes bound by PCGF6-PRC1. Expression levels of the indicated genes in mock-transfected *Ring1A$^{-/-}$;Ring1B$^{fl/fl}$;Rosa26::CreERT2$^{tg/+}$* ESCs before (*Ring1A$^{-/-}$*) and 2 day after OHT treatment (*Ring1A/B*-dKO) and in those stably expressing wild type (WT) or catalytically-dead (I53S, I53A) Ring1B. Expression levels were normalized to a *Gapdh* control and are depicted as fold change relative to OHT-untreated mock-transfected parental ESCs (*Ring1A$^{-/-}$*). Those in wild-type ESCs (WT) are also shown. Error bars represent standard deviation determined from at least three independent experiments.

The following figure supplement is available for figure 2:

**Figure supplement 1.** Repression of target genes by PCGF6-PRC1.

suggested to contribute to RING1B-mediated repression of target genes primarily in the PCGF6 +RING1B+ category but also in RING1B targets not bound by PCGF6 to a lesser extent. We thus suggest PCGF6+RING1B+ category as a primary target for PCGF6-PRC1-mediated gene repression.

## PCGF6 represses germ cell-related genes via RING1B catalytic activity

To gain insight into the physiological role of PCGF6-PRC1-mediated gene regulation in ESCs, we performed gene ontology (GO) analyses, which revealed that PCGF6-PRC1 target genes are enriched in meiosis- and germ cell-related genes as suggested previously (*Zdzieblo et al., 2014*), unlike the cPRC1 target genes that are predominantly enriched in developmental genes (*Figure 2— figure supplement 1D*). These results suggest a potential role of PCGF6-PRC1 for suppression of the meiotic program or germ cell differentiation program, or both, in ESCs.

The RING1A/B proteins play a key role for transcriptional repression by mediating H2AK119ub1 (*Endoh et al., 2012*; *Wang et al., 2004*). We therefore investigated whether PCGF6-PRC1-mediated gene regulation involves H2AK119ub1. Indeed, ChIP-qPCR analyses revealed considerable enrichment of H2AK119ub1 at PCGF6-PRC1 target genes (*Figure 2C*). Furthermore, this H2AK119ub1 enrichment was dramatically diminished in the absence of *Pcgf6*, revealing a role for PCGF6 in mediating H2AK119ub1 deposition (*Figure 2C*). In contrast, H2AK119ub1 enrichment at *Zic1* and *Hoxd11,* cPRC1 targets, was marginally affected in *Pcgf6*-KO ESCs, further supporting our model that PCGF6 is not primarily involved in cPRC1-dependent repression (*Figure 2C*, middle).

To gain insight into the functional involvement of PCGF6-mediated H2AK119ub1 in repression of the target genes, we next examined the effect of loss of RING1B catalytic activity on expression of these genes (*Figure 2D*). Previous reports have demonstrated that mutations of RING1B residue I53 (i.e., I53S or I53A), abrogated its E3 ligase activity by disrupting its interactions with the E2 ubiquitin-conjugating enzyme UBCH5C (also known as UBE2D3: Ubiquitin-conjugating enzyme E2 D3) (*Ben-Saadon et al., 2006*; *Buchwald et al., 2006*; *Endoh et al., 2012*; *Eskeland et al., 2010*). Rescuing the *Ring1A/B*-dKO cells with the I53 mutants failed to maintain transcriptional repression, demonstrating that repression of PCGF6 target genes depends on the catalytic activity of RING1B and likely also downstream H2AK119ub1 deposition (*Figure 2D*).

Since our previous report demonstrated that H2AK119ub1 plays a critical role for recruitment of PRC2 and downstream H3K27me3 deposition (*Blackledge et al., 2014*), we thus investigated H3K27me3 deposition at the representative PCGF6 bound genes, and indeed found a significant H3K27me3 enrichment at these targets (*Figure 2—figure supplement 1E*). As expected, the H3K27me3 level was markedly reduced in the absence of the key PRC2 component EED (embryonic ectoderm development), confirming that H3K27me3 deposition in these genes is mediated by the PRC2 complex (*Figure 2—figure supplement 1E*). We indeed found a considerable reduction in H3K27me3 level at these PCGF6-PRC1 targets in *Pcgf6*-KO ESCs, but only marginally at cPRC1 targets, suggesting a potential role for PCGF6-mediated H2AK119ub1 to recruit PRC2 (*Figure 2C*). However, consistent with a previous report (*Riising et al., 2014*), PRC2-mediated mechanisms exhibited only limited impact on the repression of PCGF6-PRC1 targets (*Figure 2—figure supplement 1F*).

# PCGF6 recruits RING1B to target genes and facilitates downstream H2AK119ub1 and H3K27me3 deposition

Previous studies have reported that PCGF6 interacts with RING1B through its RING finger domain (*Akasaka et al., 2002*). Given that PCGF6 loss led to a marked reduction in H2AK119ub1 deposition specifically at PCGF6-PRC1 target genes (*Figure 2C*), we wondered whether PCGF6 is directly involved in recruitment of RING1B and its catalytic activity. To test this model, we expressed a PCGF6 protein fused to the TET repressor (PCGF6/TETR), and ectopically tethered this fusion protein to a pre-integrated TetO array in ESCs (*Figure 3A*), taking advantage of an experimental system that we have previously described (*Blackledge et al., 2014*). ChIP-qPCR analysis showed binding of the PCGF6/TETR fusion protein at the TetO array. Importantly, this was accompanied by considerable enrichment of RING1B, MAX and H2AK119ub1, indicating that PCGF6 recruits PCGF6-PRC1 (*Figure 3A*). We also observed association of EZH2 and H3K27me3 with the TetO array, likely mediated by the upstream H2AK119ub1 as reported previously (*Blackledge et al., 2014*).

To confirm these in vitro findings in vivo, we performed RING1B ChIP-seq in WT and *Pcgf6*-KO ESCs. This experiment demonstrated that RING1B binding to PCGF6-PRC1 target genes such as *Tex19.1*, *Tdrkh*, *Syce1* (synaptonemal complex central element protein 1) and *Mael* (maelstrom spermatogenic transposon silencer) was indeed considerably depleted in the absence of *Pcgf6* (*Figure 3B*). In contrast, RING1B binding to cPRC1 target sites, such as the *Hoxb* cluster, was unaffected.

To determine the role of PCGF6 in RING1B recruitment at ncPRC1 targets in a comprehensive manner, we divided RING1B-bound genes into four categories (*Figure 3C*). The first category consisted of genes bound by RING1B but not PCGF6 or CBX7 (PCGF6-CBX7-RING1B+). Genes in the second category were bound by both PCGF6 and RING1B, but not CBX7 (PCGF6+CBX7-RING1B+); and therefore represented the core targets of the non-canonical PCGF6-PRC1 complex. The third category consisted of the cPRC1 target genes, which were bound by CBX7 and RING1B, but not PCGF6 (PCGF6-CBX7+RING1B+). Finally, the fourth category was marked by all three factors (PCGF6+CBX7+RING1B+). We observed considerable reduction of RING1B binding to the core PCGF6-PRC1 target genes (PCGF6+CBX7-RING1B+) in *Pcgf6*-KO (green dot plots in *Figure 3C*, left; green lines in *Figure 3C*, right). In contrast, cPRC1 target genes (PCGF6-CBX7+RING1B+) were not affected. Consistent with these observations, genetic ablation of *Pcgf2* (*Mel18*) and *Pcgf4* (*Bmi1*), both of which are essential components of the cPRC1 complex, led to reduction of RING1B binding of only cPRC1 target genes (PCGF6-CBX7+RING1B+) (blue dot plots in *Figure 3C*, left; blue lines in *Figure 3C*, right). These observations were further confirmed by ChIP-qPCR for selected target genes (*Figure 3—figure supplement 1A*). Taken together, these findings indicate that PCGF6 preferentially recruits RING1B to PCGF6-PRC1 targets, but not to cPRC1 targets.

To determine whether PCGF6-dependent RING1B recruitment involved direct physical interactions between these two proteins, we tested a PCGF6 mutant with an amino acid substitution (H155Y) in a conserved residue critical for interaction with RING1B (*Buchwald et al., 2006*). We expressed the mutant in *Pcgf6*-KO ESCs, and noted that it indeed failed to interact with RING1B, while retaining the ability to associate with L3MBTL2 and MAX (*Figure 3—figure supplement 1B*). ChIP-qPCR analysis showed that although the mutant PCGF6 could bind the indicated target genes, it failed to recruit RING1B (*Figure 3D*) and, as a result, H2AK119ub1 and H3K27me3 marks were also not established in the target genes. Consistent with this interpretation, the mutant PCGF6 failed to repress PCGF6-PRC1 targets genes (*Figure 3—figure supplement 1C*).

Based on these observations, we concluded that PCGF6 contributes to the assembly and recruitment of the PCGF6-PRC1 complex by direct physical interactions with RING1B, which likely plays a critical role to mediate the repression. Previous and these studies, however, suggest potential involvement of other mechanisms, such as L3MBTL2, G9A/GLP and CBX1/3 for the repression by PCGF6-PRC1 (*Maeda et al., 2013*; *Qin et al., 2012*; *Trojer et al., 2011*). Indeed, we noted that PCGF6 was dispensable for L3MBTL2 binding of gene promoters (*Figure 3E*). As L3MBTL2 has been reported as a recruiter of ncPRC1 (*Qin et al., 2012*; *Trojer et al., 2011*), this observation suggests that PCGF6 and L3MBTL2 may function in parallel to recruit ncPRC1 complexes. We went on to examine contribution of H3K9me2, which is mediated by G9A/GLP, for PCGF6-PRC1-target repression (*Figure 3—figure supplement 1D*). Unlike *L3mbtl2*-KO ESCs (*Qin et al., 2012*), we did not observe considerable changes in H3K9me2 levels in *Pcgf6*-KO. Together, we suggest that the

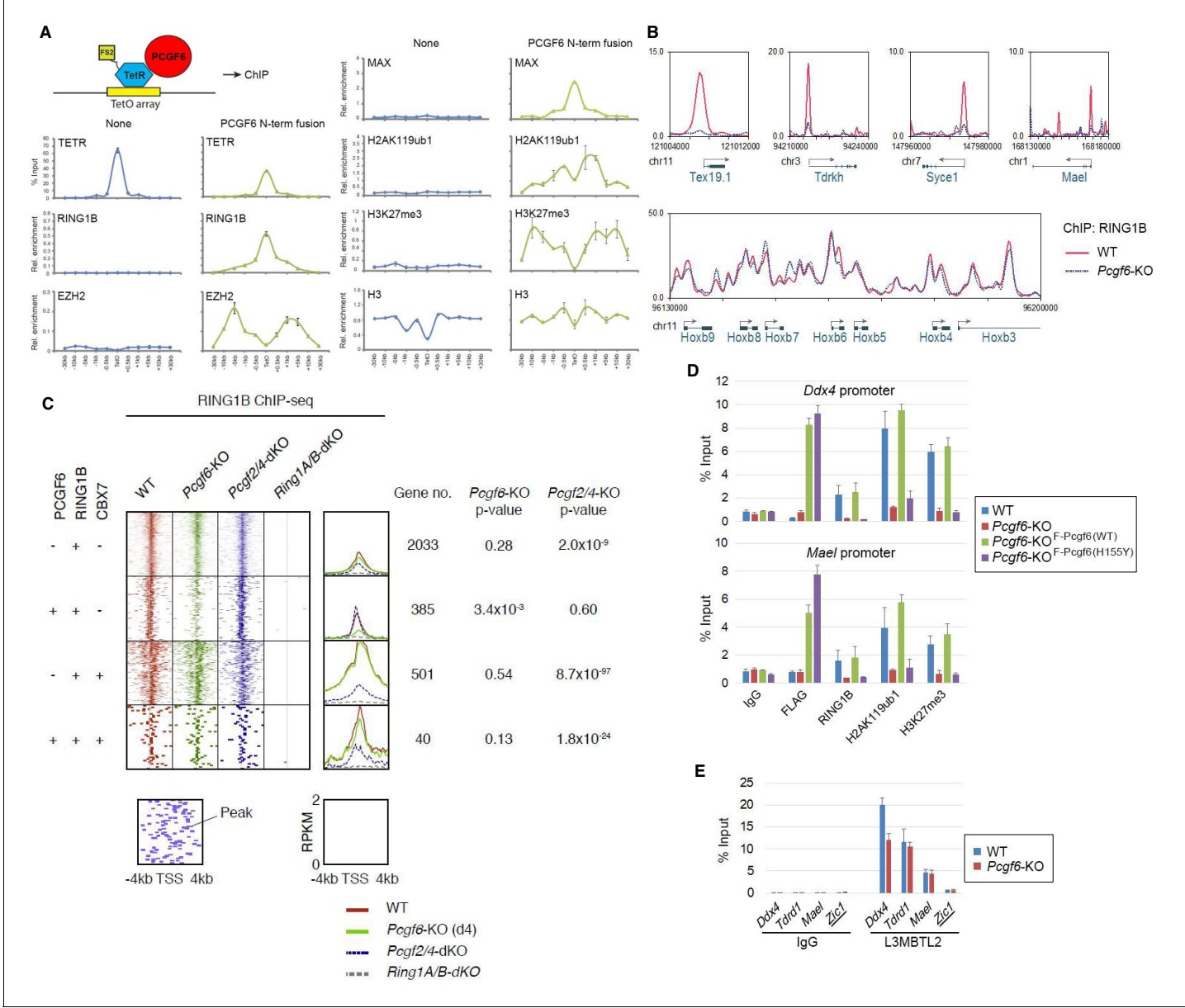

**Figure 3.** The role of PCGF6 in recruiting PCGF6-PRC1 to its target genes. (A) Forced tethering of PCGF6 to a TetO array induced activation of PCGF6-PRC1 recruitment. A schematic illustrating the de novo targeting of the TetR-PCGF6 fusion protein to the TetO sequences (left). ChIP analysis for TetR, RING1B, EZH2, MAX, H2AK119ub1, H3K27me3 and H3 across the TetO-containing locus in ESCs revealed TetR-PCGF6-mediated local activation of the PCGF6-PRC1 pathway (bottom). ChIP experiments were performed at least in biological duplicate with error bars showing SEM. (B) Screenshot views of the RING1B distribution at selected target genes in wild type (WT) and *Pcgf6*-KO ESCs. RING1B distribution in *Pcgf6^fl/fl^;Rosa26::CreERT2^tg/+^* ESCs before (WT; red) or 4 days after OHT treatment (*Pcgf6*-KO; blue) revealed by ChIP-seq is shown. The chromosome numbers and locations are indicated on the x-axis. The transcription start sites (TSSs) are denoted by arrows. (C) A heat map view of RING1B distribution in ±4 kb genomic regions around TSS in wild type (WT; brown), *Pcgf6*-KO (green), *Mel18/Bmi1*-dKO (blue) and *Ring1A/B*-dKO (gray) ESCs. RING1B-bound genes are further subclassified by binding of PCGF6 and CBX7 (left). Average distribution and p-value for its change in respective KO are also shown (right). (D) PCGF6-mediated RING1B recruitment to its target via a direct molecular interaction. Local levels of Flag-tagged PCGF6, RING1B, H2AK119ub1 and H3K27me3 at *Ddx4* and *Mael* promoters in mock-transfected *Pcgf6^fl/fl^;Rosa26::CreERT2^tg/+^* ESCs before (WT) and after OHT treatment (*Pcgf6*-KO) and in ESCs stably expressing WT [*Pcgf6*-KO_F-Pcgf6(WT)] or H155Y Pcgf6 [*Pcgf6*-KO_F-Pcgf6(HY)] constructs were determined by ChIP-qPCR. The relative amount of ChIPed DNA is depicted as a percentage of input DNA. Error bars represent standard deviation determined from at least three independent experiments. (E) PCGF6 is dispensable for local L3MBTL2 binding to target genes. Local levels of L3MBTL2 at the promoter regions of the indicated genes in *Pcgf6^fl/fl^;Rosa26::CreERT2^tg/+^* ESCs before (WT) or 4 days after OHT treatment (*Pcgf6*-KO) were determined by ChIP-qPCR. Underlined genes are canonical PRC1 targets.

*Figure 3 continued on next page*

*Figure 3 continued*

The following figure supplement is available for figure 3:

**Figure supplement 1.** The role of PCGF6 in recruiting PCGF6-PRC1 to its target genes.

L3MBTL2/G9A axis is marginally involved in PCGF6-mediated repression. We further investigated the role of CBX1/3 for the regulation of PCGF6-PRC1 targets by using tamoxifen-inducible *Cbx1* and *Cbx3* double knockout (*Cbx1/3*-dKO) ESCs (*Figure 3—figure supplement 1E*). We did not observe a significant change in the expression levels of PCGF6+CBX7-RING1B+ in *Cbx1/3*-dKO ESCs, suggesting that CBX1/3 have, if any, a marginal role for the repression of PCGF6-PRC1 targets in ESCs (*Figure 3—figure supplement 1F*).

## The MAX/MGA heterodimer is required for recruitment of PCGF6-PRC1 to target genes

The above results establish a role for PCGF6 in the recruitment of RING1B, however, the mechanism by which PCGF6-PRC1 is targeted to such loci is not understood. According to the canonical model, H3K27me3 facilitates PRC1 binding via direct recruitment of CBX proteins (*Bernstein et al., 2006*; *Cao et al., 2002*; *Czermin et al., 2002*; *Fischle et al., 2003*; *Kuzmichev et al., 2002*); we therefore examined whether H3K27me3 plays a role in PCGF6 recruitment (*Figure 4—figure supplement 1A, B*). For this purpose, we stably expressed a FLAG-tagged PCGF6 in WT or *Eed*-KO ESCs (deficient for H3K27me3), and compared PCGF6-binding levels at target genes. We did not observe any significant differences in FLAG-PCGF6 binding between WT and *Eed*-KO cells (*Figure 4—figure supplement 1B*), suggesting that H3K27me3 is dispensable for locus-specific recruitment of PCGF6-PRC1. Consistently, H2AK119ub1 levels at genes bound by PCGF6 were not affected in the *Eed*-KO (*Figure 4—figure supplement 1C*). In contrast, PCGF2, a component of cPRC1, barely associates with PCGF6 targets irrespective of H3K27me3 status, while it strongly associates with cPRC1 targets such as *Zic1* and *Hoxb3* in an EED-dependent manner (*Figure 4—figure supplement 1C*). These findings suggests that the H3K27me3-dependent pathway to recruit cPRC1 is barely active at PCGF6-PRC1 targets.

To determine the molecular mechanism that mediates recruitment of PCGF6-PRC1 to target loci, we surveyed the DNA sequences of PCGF6-bound promoters to extract potential transcription factor binding motifs. We observed enrichment of the E-box motif CACGTG, a DNA sequence that is recognized by the bHLH-containing transcription factors including MAX (*Blackwood and Eisenman, 1991*), within these promoters (*Figure 4—figure supplement 1D*). Importantly, this motif was not enriched in RING1B- or CBX7-bound genes (*Figure 4—figure supplement 1D*), suggesting that the E-box motif is a feature of PCGF6-PRC1 targets. Consistent with this notion, we observed considerable overlap between PCGF6-bound and MAX-bound genes, while the overlap between PCGF6-bound and MYC-bound genes was much less (*Figure 4A*). Interestingly, MGA, a transcription factor that also forms a heterodimer with MAX and binds the CACGTG E-box motif (*Hurlin et al., 1999*), is included in the PCGF6 complex determined by us (*Figure 1B*) and others (*Gao et al., 2012*; *Hauri et al., 2016*; *Kloet et al., 2016*; *Ogawa et al., 2002*; *Qin et al., 2012*; *Trojer et al., 2011*). Therefore, we hypothesized that MAX/MGA heterodimer could play a role in PCGF6-PRC1 recruitment.

To examine this possibility, we depleted MAX or MGA by siRNA-mediated knockdown (KD) in FLAG-PCGF6 expressing ESCs. The *Max* or *Mga* mRNA levels were reduced to less than 20% of control by their respective siRNAs (*Figure 4—figure supplement 1E*, top). Western blot analysis revealed that MAX protein was undetectable in the *Max*-KD ESCs, while the expression of FLAG-PCGF6 and RING1B was unchanged (*Figure 4—figure supplement 1E*, bottom). To determine the impact of MAX or MGA depletion, we performed RNA-seq in *Max*-KD or *Mga*-KD ESCs. Interestingly, genes that were bound by MAX, PCGF6 and RING1B (MAX+PCGF6+RING1B+) showed significant up-regulation in *Max*-KD or *Mga*-KD ESCs (*Figure 4B*). We performed locus-specific RT-qPCR assays in the *Ddx4* [DEAD (Asp-Glu-Ala-Asp) box polypeptide 4], *Tdrd1* (Tudor domain containing 1), *Stag3* (stromal antigen 3) and *Mael* loci, and confirmed our RNA-seq results (*Figure 4—figure supplement 1F*).

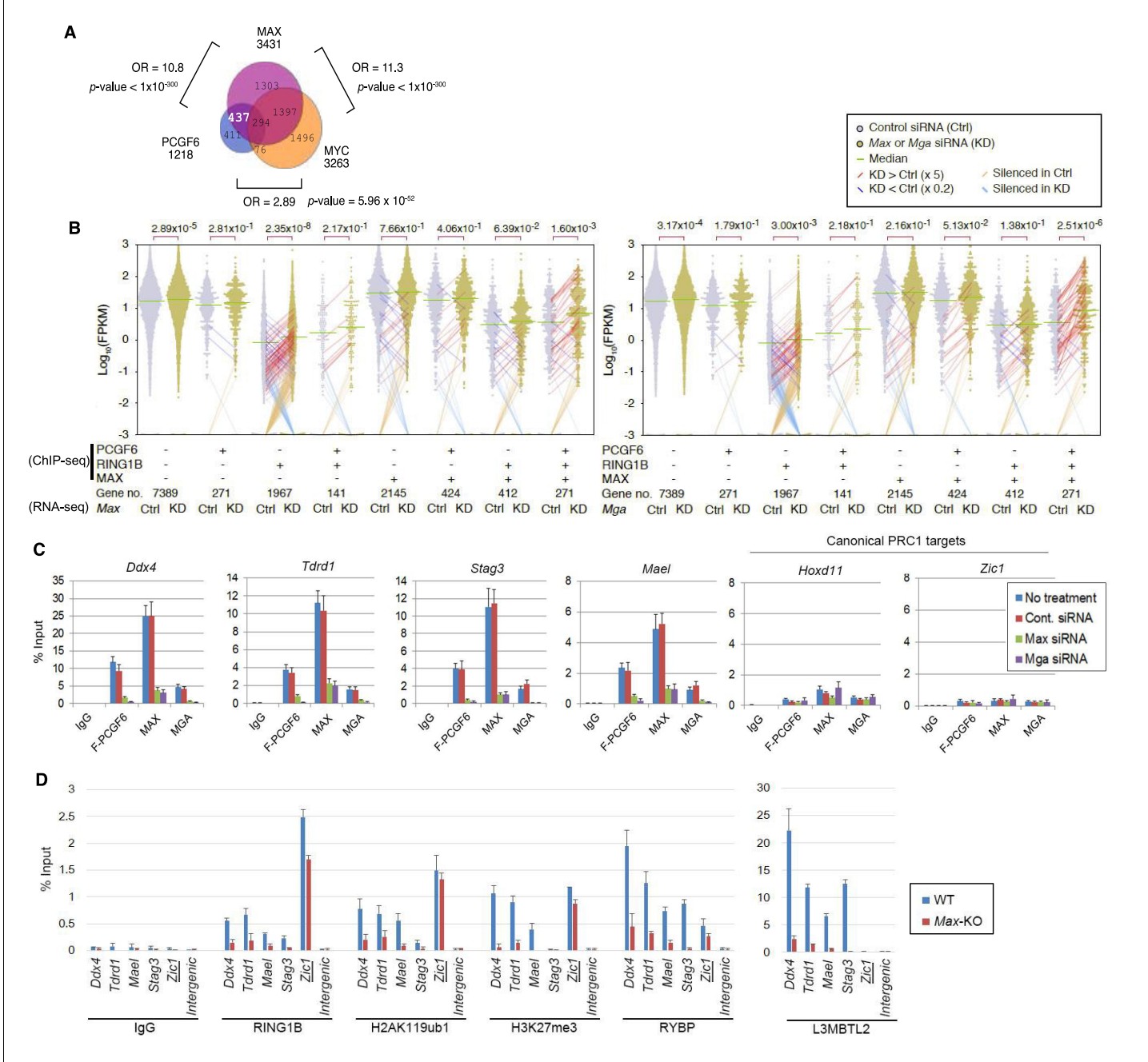

**Figure 4.** The role of MAX/MGA in recruiting PCGF6-PRC1 to its target genes. (**A**) Considerable overlap of genes bound by PCGF6 and MAX. Venn diagram depicts the overlap of PCGF6, MAX and MYC target genes in ESCs. Published ChIP-seq data for MAX and MYC were obtained from NCBI GEO (accession number GSM1171650 and GSM1171648, respectively). Odds ratio (OR) and *p*-values for the overlap between respective groups are indicated. Numbers represent the number of genes bound by each protein or included in each fraction seen in the Venn diagram. (**B**) Dot plot representation for gene expression changes in *Max*-KD (left) and *Mga*-KD (right) ESCs. Expression levels of respective genes (FPKM: determined by RNA-seq) in each group (classified by ChIP-seq) in control (Ctrl) and respective knockdowns (KD) are shown by grey and yellow dots, respectively. The same codes as described in *Figure 2A* are used. (**C**) Changes in binding of FLAG-tagged PCGF6, MAX and MGA at the selected targets induced by knockdown of *Max* or *Mga*. Local levels of FLAG-tagged PCGF6, MAX and MGA at the respective promoter regions in untreated ESCs or ESCs treated with either control siRNA, *Max* siRNA or *Mga* siRNA were determined by ChIP-qPCR. The relative amount of ChIPed DNA is depicted as a percentage of input DNA. Error bars represent standard deviation determined from at least three independent experiments. (**D**) Changes in local deposition of RING1B, H2AK119ub1, H3K27me3, RYBP and L3MBTL2 at the indicated targets in *Max*-KO ESCs. Underlined genes are canonical PRC1 targets. Their deposition in *Max* conditional KO ESCs before (WT) or after doxycycline treatment (*Max*-KO) was determined by ChIP-qPCR. The relative amount of

*Figure 4 continued on next page*

*Figure 4 continued*

ChIPed DNA is depicted as a percentage of input DNA. Error bars represent standard deviation determined from at least three independent experiments.

The following figure supplements are available for figure 4:

**Figure supplement 1.** The role of MAX/MGA in recruiting PCGF6-PRC1 to its target genes.

**Figure supplement 2.** Sequence recognition by MAX/MGA is critical for recruiting PCGF6-PRC1 to its target genes.

In contrast, genes bound by PCGF6 and RING1B, but not MAX (MAX-PCGF6+RING1B+), were barely up-regulated (*Figure 4B*). Intriguingly, we observed significant up-regulation of genes bound only by RING1B (MAX-PCGF6-RING1B+). We however also noted that the expression level of up-regulated genes in this category was considerably lower compared to MAX+PCGF6+RING1B+ genes (*Figure 4B*). We thus suggest MAX and MGA primarily contribute to the repression of MAX+PCGF6+RING1B+ genes. GO analysis revealed that genes bound by PCGF6, RING1B and MAX but not by MYC (PCGF6+RING1B+MAX+MYC-), were significantly enriched in germ cell- and meiosis-related functions (*Figure 4—figure supplement 1G*), further supporting our model that PCGF6-PRC1 functions as a dedicated repressor of genes associated with germ cells and meiosis.

We investigated whether binding of PCGF6-PRC1 to gene promoters required MAX and its cofactor MGA. Indeed, ChIP-qPCR in the *Max*-KD or *Mga*-KD ESCs revealed that PCGF6 binding to target loci depends on MAX and MGA (*Figure 4C*). Importantly, MAX and MGA bind these targets in mutually dependent manner (*Figure 4C*). This suggests that MAX and MGA may form a dimer to bind to their targets. We further used a previously reported *Max*-KO ES cell line to test MAX-dependent recruitment of PCGF6-PRC1 (*Hishida et al., 2011*). RING1B and RYBP, which form stoichiometric complexes with PCGF6 (*Figure 1B*), did not bind to PCGF6-PRC1 target loci in the *Max*-KO ESCs (*Figure 4D*). Likewise, local H2AK119ub1 and H3K27me3 levels were considerably decreased, and binding of L3MBTL2, another key regulator of ncPRC1, into PCGF6-PRC1 target genes was also depleted in the *Max*-KO ESCs (*Figure 4D*). In contrast, enrichment of RING1B, H2AK119ub1, H3K27me3 and RYBP at *Zic1*, a cPRC1 target, did not show considerable changes in the *Max*-KO ESCs. These results suggest that recruitment of both PCGF6 and L3MBTL2 requires MAX. We finally checked whether the decreased binding of PCGF6 is not due to secondary effects of transcriptional activation upon MAX depletion. We have selected genes bound by MAX and PCGF6, which were not up-regulated in *Max*-KO ESCs, and confirmed MAX-dependent binding of PCGF6 at these targets (*Figure 4—figure supplement 1H*). Collectively, these findings indicate that the MAX/MGA heterodimer recruits PCGF6-PRC1 to its targets.

## Interactions between MAX and PCGF6 play a role in ESC maintenance

A role for MAX in maintaining ESC proliferation and self-renewal ability has been reported in a previous study (*Hishida et al., 2011*). We also observed that conditional deletion of *Max* (*Max*-KO) led to slower ESC proliferation and cell differentiation (*Figure 4—figure supplement 2A*, right). Rescuing the induced *Max*-KO cells with an exogenous WT MAX restored cell proliferation and blocked spontaneous differentiation, as expected. In contrast, mutant MAX proteins harboring defects in the basic region of the bHLH (basic helix-loop-helix) domain [hereafter referred to as MAX(VD) or MAX(Δb)] (*Figure 4—figure supplement 2A*, left), failed to do so (*Figure 5A*, right). These findings indicate that DNA motif recognition by MAX plays a role in maintaining ESCs, at least partly by recruiting the PCGF6-PRC1 complex.

Mechanistically, co-IP analyses showed that a FLAG-tagged MAX protein physically interacts with HA-PCGF6, RING1B and L3MBTL2. The mutant MAX proteins also retained the ability to associate and form stable complexes with HA-PCGF6, RING1B and L3MBTL2 (*Figure 4—figure supplement 2B*). As expected, ChIP-qPCR revealed that the FLAG-tagged MAX could restore binding of the HA-tagged PCGF6 into target loci in *Max*-KO ESCs (*Figure 4—figure supplement 2C,D*), whereas the mutant MAX proteins failed to bind targets and recruit HA-tagged PCGF6. These observations demonstrate a role of the MAX/MGA heterodimer in recruiting PCGF6-PRC1 by direct DNA motif

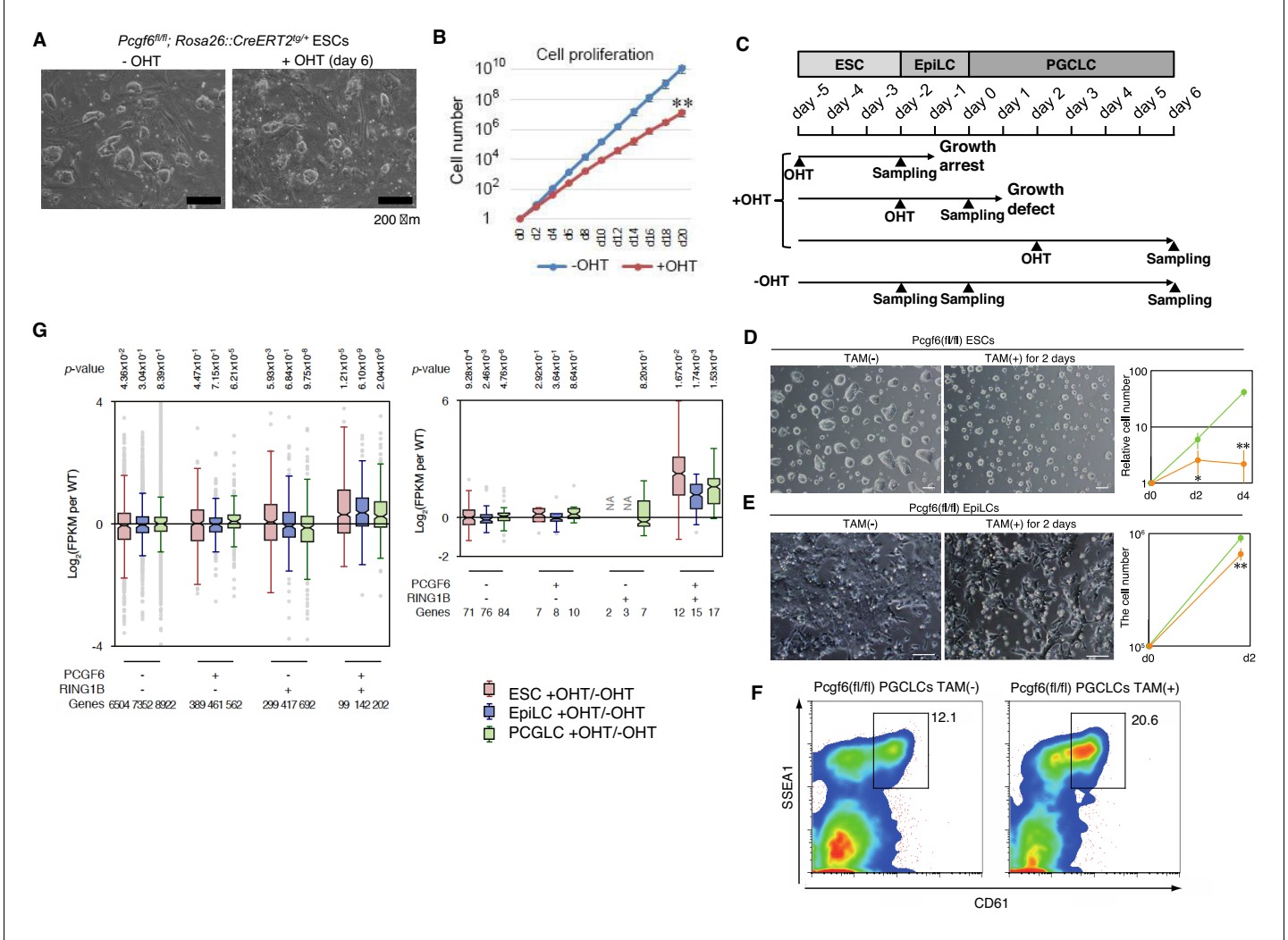

**Figure 5.** A role for PCGF6 in maintaining ESCs in an undifferentiated state. (**A**) Colony morphology of *Pcgf6*-KO ESCs in the presence of fetal bovine serum (FBS) and 3i. Phase-contrast views of OHT-untreated and –treated (day six) *Pcgf6^{fl/fl}*;*Rosa26::CreERT2^{tg/+}* ESCs are shown. Scale bars indicate 200 μm. (**B**) Decreased growth rate of *Pcgf6*-KO ESCs. Proliferation of OHT-untreated or -treated *Pcgf6^{fl/fl}*;*Rosa26::CreERT2^{tg/+}* ESCs in the presence of FBS and 3i is shown. (**C**) A schematic representation of the in vitro differentiation scheme for *Pcgf6^{fl/fl}*;*Rosa26::CreERT2^{tg/+}* ESCs towards EpiLCs and PGCLCs is shown. OHT was added to deplete *Pcgf6* at the indicated time points (closed arrowheads with 'OHT'). Cells used for RNA-seq analysis were collected at the time points indicated as 'Sampling'. (**D**) PCGF6 is indispensable to maintain proliferation of ESCs in serum-free condition. Phase contrast views of *Pcgf6*-KO [TAM(+); at two days after OHT treatment] and the control [TAM(−)] (left). Growth rates of OHT-treated (yellow) and –untreated (green) cells (right). (**E**) PCGF6 is indispensable to maintain proliferation of epiLCs. Phase contrast views of *Pcgf6*-KO [TAM(+); at 2 days after OHT treatment] and the control [TAM(−)] (left). Growth rates of OHT-treated (yellow) and –untreated (green) cells (right). (**F**) PCGF6 is dispensable for differentiation of post-epiLCs towards PGCLC. OHT-treatment considerably expanded the SSEA1+CD61+ fraction. (**G**) Gene expression changes upon induced deletion of *Pcgf6*. *Pcgf6^{fl/fl}*;*Rosa26::CreERT2^{tg/+}* ESCs (red), EpiLCs (blue), and PGCLCs (green) induced upon depletion of *Pcgf6* by OHT treatment for each subset of total genes (left) or meiosis-related genes (right) classified by the presence (+) or absence (−) of PCGF6- and RING1B-binding in ESCs. The average, deviation and distribution of the expression changes for the respective subsets of genes determined by RNA-seq analysis are shown. The box plots represent the median (horizontal line), interquartile range (box), range (whiskers), and outliers (circles). The number of genes included in each subset is shown at the bottom. *p*-values for average gene expression change in each subset upon Pcgf6 depletion were calculated by the Student's *t*-test and are indicated at the top.

recognition for maintenance of ESCs. To further examine this model, we expressed MAX/TETR, and ectopically tethered this fusion protein to a pre-integrated TetO array in ESCs (*Figure 4—figure supplement 2E*). ChIP-qPCR analysis showed binding of the MAX/TETR fusion protein at the TetO array, which was accompanied by mild enrichment of HA-tagged PCGF6, H2AK119ub1 and

H3K27me3 (*Figure 4—figure supplement 2E*), but barely RING1B (data not shown). These data again support MAX-dependent recruitment of PCGF6-PRC1. The limited effect of MAX/TETR to recruit PCGF6 could be due to a potential dimerization of MAX with other partners than MGA in this experimental setup.

## PCGF6-PRC1 is required for maintenance of ESCs and epiLCs in an undifferentiated state

Given that MAX and PCGF6 interact with each other, and that genetic ablation of *Max* causes reduced cell growth and germ cell-directed differentiation of ESCs (*Hishida et al., 2011*; *Suzuki et al., 2016*), we wondered whether knocking out *Pcgf6* would also lead to similar phenotypes. Indeed, the growth rate of *Pcgf6*-KO ESCs was significantly slower than the WT in the presence of three inhibitors (3i: SU5402 for FGFR, PD184352 for ERK, and CHIR99021 for GSK3) and fetal bovine serum as suggested previously (*Zdzieblo et al., 2014*) (*Figure 5A,B*). The growth of *Pcgf6*-KO ESCs was further halted under serum-free condition (*Figure 5D*). Unlike the *Max*-KO, however, *Pcgf6*-KO ESCs maintained an ESC-like morphology. These findings support the notion that PCGF6-PRC1 has a role in ESC proliferation, but may not be essential to maintain ES cell morphology.

Genetic ablation of *Pcgf6* leads to ectopic expression of meiosis-related and germ cell-related genes in ESCs. We therefore examined the impact of *Pcgf6* deletion in primordial germ cell-like cells (PGCLCs), which already express such germ cell-related genes. We induced the PGCLCs via epiblast-like cells (epiLCs), using an in vitro culture system that we have previously developed (*Hayashi et al., 2011*). In particular, we cultured the ESCs in 2i-containing (MAPK inhibitor and GSK3 inhibitor) media without feeders for 3 days (from day −5 to day −2 in *Figure 5C*), and induced epiLC by adding Activin A and bFGF into the media for 2 days (from day −2 to day 0). We further differentiated the epiLCs into the germ cell-lineage by adding BMPs, SCF, LIF, and EGF for 6 days (from day 0 to day 6). Genetic ablation of *Pcgf6* at day −5 induced growth arrest and cell death of ESCs (*Figure 5D*); while ablation of *Pcgf6* at day −2 caused significant growth defects in epiLCs (*Figure 5E*), revealing a stage-specific role of PCGF6 for growth and survival of ESCs and epiLCs. In contrast, OHT-treatment at day +2 (post-epiLC stage) did not induce growth arrest or cell death and, surprisingly, even accelerated differentiation and proliferation (*Figure 5F*).

The disparate effects of *Pcgf6* ablation in ESCs, epiLCs and PGCLCs raised the question of whether the PCGF6 targets in these cell types are different. We therefore performed RNA-seq before and after knocking out *Pcgf6* in ESCs, epiLCs and PGCLCs (*Figure 5G*). We observed that the same group of genes (the PCGF6+RING1B+ category in ESCs) were de-repressed in ESCs, epiLCs and PGCLCs in the absence of *Pcgf6*. Furthermore, the extent of derepression of PCGF6 target genes was also similar among these three cell types. We observed similar trends for meiosis- and germ cell-related genes (*Figure 6G*, right). Thus, these results indicate that the different phenotypic impacts of *Pcgf6* ablation in ESCs, epiLCs and PGCLCs are not due to up-regulation of different sets of genes, but rather suggest that PCGF6-PRC1-mediated gene silencing is essential to maintain ESC and epiLC, but not PGCLC. PCGF6, therefore, plays a role in maintaining pluripotency, likely by suppressing germ cell potential in ESCs.

## A role of PCGF6 for pre- and peri-implantation embryonic development

As PCGF6 plays a role in pluripotency and suppression of germ cell-specific genes in ESCs, we wondered whether PCGF6 possessed similar functions in vivo during mouse embryogenesis. Consistently, *Pcgf6* is expressed in broad range of tissues and cell types and even abundantly in testis, ESCs and pre-implantation embryos (*Figure 6—figure supplement 1A,B*). To examine this possibility, we used a constitutive mutant *Pcgf6* allele (*Figure 6A,B,C*). We found that *Pcgf6*-KO (−/−) mice were viable and fertile, but were not born at the normal Mendelian ratio (*Figure 6D*). Lethality among $Pcgf6^{-/-}$ homozygous embryos could be observed as early as the blastocyst stage [3.5 days post coitum (dpc)]. Furthermore, embryonic death in the $Pcgf6^{-/-}$ line could be continuously detected during post-implantation development. Consistently, we found that about one third of the surviving $Pcgf6^{-/-}$ embryos at 10.5 dpc exhibited growth retardation (*Figure 6E*).

To determine the impact of P*cgf6* ablation during the pre-implantation stage, we examined WT and $Pcgf6^{-/-}$ ICMs (inner cell mass). We mated homozygous or heterozygous males with

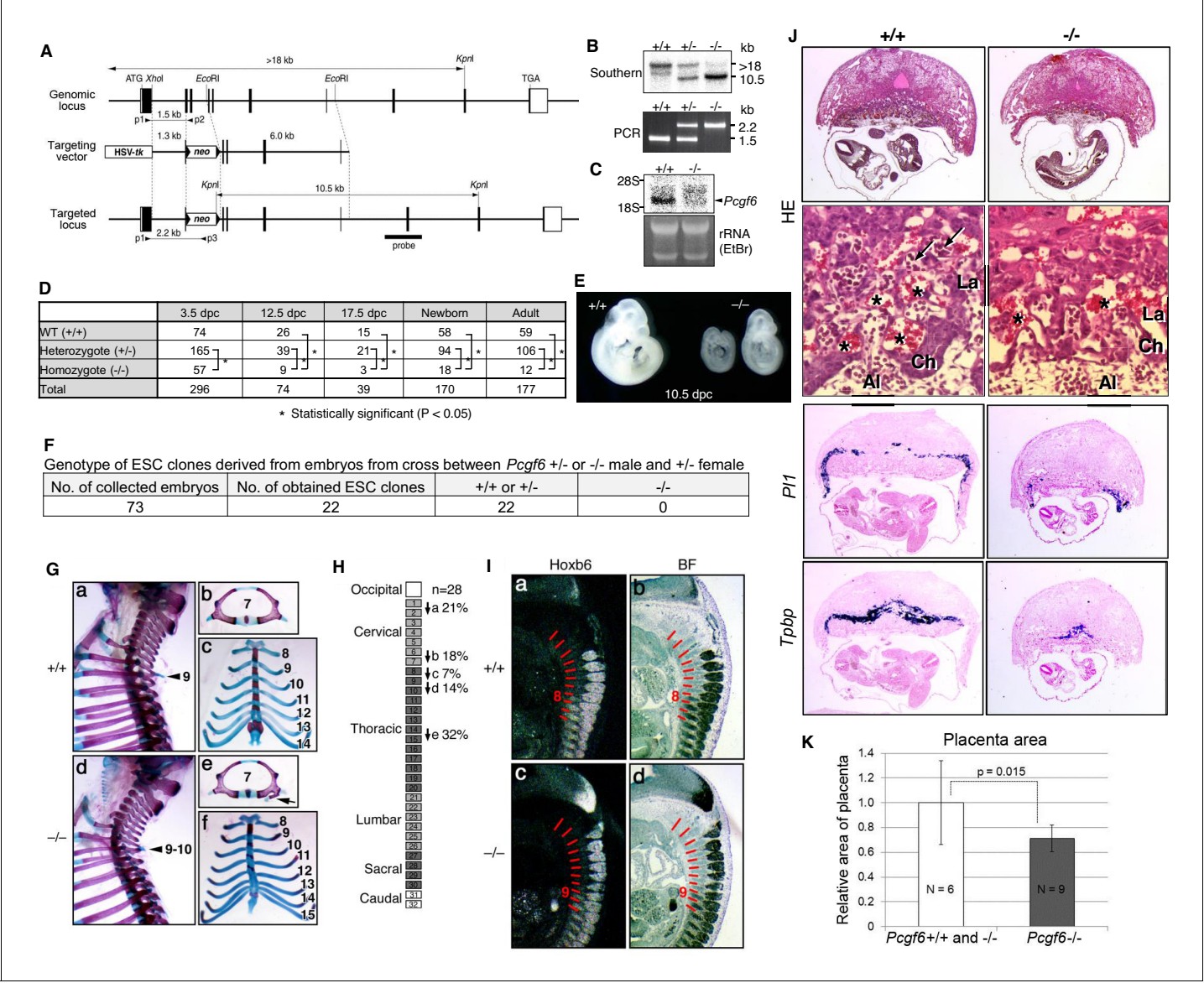

**Figure 6.** Pleiotropic effects of PCGF6 loss during development. (**A**) Schematic representation of the *Pcgf6* locus, the targeting vector, and the constitutive mutant allele. To disrupt *Pcgf6*, most of the exon 2 and the entire exon 3 which code for the RING finger domain were replaced with neo. The neo and HSV-tk cassettes were used for positive and negative selection, respectively. The positions of the restriction sites (*Xho*I, *EcoR*I and *Kpn*I), external probe and PCR primers, and sizes of diagnostic fragments are indicated. Coding regions and untranslated regions of *Pcgf6* are indicated by closed and open boxes, respectively. (**B**) Southern (top) and PCR (bottom) analyses for genotyping. For Southern blotting, genomic DNA was digested by *Kpn*I and probed with the 3' probe, as indicated in **A**. For PCR, a mixture of three primers (p1, p2 and p3 in **A**) was used. (**C**) Northern analysis of *Pcgf6* mRNA expression in wild type (+/+) and homozygous (−/−) adult kidney (top). Ethidium bromide (EtBr) staining of the same gel is shown below. (**D**) Genotype distribution of progeny of *Pcgf6* heterozygous intercrosses. *p-value<0.05 (χ² test), indicating a < 5% probability of conforming to the Mendelian law, which predicts a 1:2:1 ratio between +/+:+/−: −/−. (**E**) Delayed and/or abnormal development of *Pcgf6*-KO (−/−) embryos at 10.5 dpc. Wild-type (+/+) embryos are shown as controls. (**F**) Genotype distribution of ESC lines derived from embryos from intercrosses between *Pcgf6*+/+ or −/− male and +/− female is shown. Genotypes of 22 ESC lines were determined using genomic PCR. (**G**) Skeletal alterations in *Pcgf6*−/− newborn mice. (**a, d**) Lateral views of the cervical and thoracic regions revealed the prominent spinous process on the 9–10 in *Pcgf6*−/− (d, indicated by an arrow) but on the nine in the wild type (a, indicated by an arrow). (**b, e**) Overviews of the seventh vertebrae identified association of an anterior tubercle on seven in *Pcgf6*−/−, which appears on six in the wild type (e, indicated by an arrow). (**c, f**) Ventral views of rib cages identified an ectopic sternal rib in *Pcgf6*−/− (f, labeled 15). (**H**) Schematic representation summarizing the axial alterations in *Pcgf6*−/− newborn mice. Each arrow represents the following morphological changes in the vertebrae; (a) 1←2: association of the anterior arch of the atlas with the atlas; (b) 6←7: the anterior tubercle(s) on 7, which are normally on 6; (c) 8←9: fusion of rib(s) on 9 to 8; (d) 9←10: the prominent spinous process on 10, which is normally on 9; (e) 14←15: fusion of ribs on 15 to the sternum. (**I**) Changes in *Hoxb6* expression in *Pcgf6*-KO mice. *Hoxb6* expression in 11.5-dpc wild-type (a) and *Pcgf6*−/− (c) embryos. Segment boundaries are indicated by red lines and segment numbers of anterior expression domains of *Hoxb6* are shown. Bright field views are shown in (b, d).

*Figure 6 continued on next page*

Figure 6 continued

(J) HE sections of 11.5 dpc wild type (+/+) and *Pcgf6*-deficient (−/−) placentae. Al: allantois, Ch: chorion, La: labyrinth layer, asterisks: enucleated erythrocytes from the mother, arrows: fetus-derived nucleated erythrocytes. Note that fetus-derived nucleated erythrocytes are observed in well-developed labyrinth layer in Wild-type mice but not in *Pcgf6*-KO mice. Expression of trophoblast lineage-restricted markers was analyzed by RNA in situ hybridization. *Pl1* identifies trophoblast giant cells. *Tpbp* identifies spongiotrophoblasts. (K) Quantification of placental area from 10.5 dpc HE sections of wild type (+/+), *Pcgf6* heterozygote (+/−) and *Pcgf6*-deficient (−/−) embryos.

The following source data and figure supplement are available for figure 6:

**Source data 1.** HE sections of placentae of wild type (+/+) and *Pcgf6*-KO (−/−) embryos used for analysis shown in *Figure 6K*.

**Figure supplement 1.** Skeletal alterations in *Pcgf6*KI/KI and *Pcgf6*/*Pcgf2* dKO newborn mice.

heterozygous females, and cultured the blastocysts in ESC inducing conditions until ESC-like colonies appeared. From 73 blastocysts collected, we could establish 22 ESC-like lines. Remarkably, among these 22 lines, not a single one was homozygous $Pcgf6^{−/−}$ (*Figure 6F*). As ESCs could not be established from $Pcgf6^{−/−}$ lines, we surmised that *Pcgf6*-deficient ICMs are functionally different from their WT counterparts. Collectively, these results reveal a role for PCGF6 in both pre- and peri-implantation development.

A role of cPRC1 in the regulation of anterior-posterior (A-P) specification of the axial skeleton is well characterized (*Akasaka et al., 1996*; *Isono et al., 2005*; *Suzuki et al., 2002*). Given the essential role of PCGF6 in the PCGF6-PRC1 complex, we therefore asked whether $Pcgf6^{−/−}$ mice also exhibited defects in A-P specification. Intriguingly, we observed morphological alterations of the vertebrae that represent anterior transformations of the axis in these mice (*Figure 6G,H*). To further confirm the role of PCGF6 for A-P patterning, we then tested the impacts of PCGF6 over-expression and, indeed, observed posterior transformations of the axis (*Figure 6—figure supplement 1C,D*). We further found that skeletal alterations in $Pcgf6^{−/−}$ mice were accompanied by aberrant repression of *Hoxb6* at the eighth pre-vertebra (*Figure 6I*). These suggest that PCGF6 regulates A-P patterning presumably through regulation of *Hox* genes. We then examined whether skeletal defects in $Pcgf6^{−/−}$ involved potential interactions of PCGF6-PRC1 with cPRC1 at *Hox* genes (*Figure 1D*, *Figure 6—figure supplement 1E,F*). We indeed observed that anterior transformations in $Pcgf6^{−/−}$ mice were suppressed by the *Pcgf2* mutation, suggesting mutually counteracting properties of PCGF6 and PCGF2.

In addition to the skeletal phenotypes, we also noted significant reduction of placental size, manifested by hypoplasticity and malformation of the labyrinth layer, in the $Pcgf6^{−/−}$ mice (*Figure 6J,K*). In particular, the number of spongiotrophoblast cells and trophoblast giant cells (TGCs), identified by expression of *Pl1* (placental lactogen-I, also known as *Prl3d1*) and *Tpbpa* (trophoblast specific protein alpha), respectively, was dramatically decreased. These findings demonstrate further roles of PCGF6 in peri-implantation development, including A-P specification and regulation of placental development.

## Discussion

### PCGF6 constitutes a non-canonical PRC1 in association with RING1B

In the present study, we show that PCGF6 constitutes a non-canonical PRC1 complex, PCGF6-PRC1, that includes the core PRC1 component RING1B. PCGF6 physically interacts with RING1B, and plays a role in recruitment of RING1B to target genes, followed by deposition of H2AK119ub1 and transcriptional silencing. We have previously reported that the PCGF1-associated ncPRC1 (PCGF1-PRC1) recruits PRC2 to target genes (*Blackledge et al., 2014*). Here, we demonstrate that PCGF6-PRC1 also has a similar role, as ectopic targeting of a PCGF6 protein fused to the Tet repressor (PCGF6/TETR) to a pre-integrated TetO array leads to sequential binding of RING1B, H2AK119ub1 deposition and PRC2 recruitment in cis (*Figure 3A*).

## PCGF6-PRC1 target genes are different from those of PCGF1-PRC1 and canonical PRC1

Like PCGF1-PRC1, PCGF6-PRC1-mediated targeting of PRC2 also takes place in endogenous genomic loci, in particular at promoters of a certain group of genes that are enriched for germ cell-specific and meiosis-specific functions (*Figure 2—figure supplement 1D*). However, these genes are distinct from those bound by PCGF1-PRC1 and canonical PRC1, which mainly include developmental genes (*Figure 1D–G*). Such developmental genes are evolutionarily conserved targets of PcG complexes, and are characterized by long CpG island (CGI) promoters that are usually not marked by DNA methylation (*Sharif et al., 2013*). The CXXC domain containing protein KDM2B recruits PCGF1-PRC1 into these loci by binding unmethylated CGIs (*Blackledge et al., 2014*). In contrast, PCGF6-PRC1 target loci tend to have slightly shorter and more DNA methylated CGIs (*Figure 1—figure supplement 1E*), where KDM2B seems to play distinct roles from PCGF1-PRC1. Indeed, H3K27me3 depositions at PCGF6-PRC1 targets are less active to recruit cPRC1 though underlying mechanisms await elucidative (*Figure 1G*, *Figure 4—figure supplement 1C*). Thus, our findings reveal that there are at least two major modes of gene silencing by PRC1; the first involves PCGF1-PRC1 and cPRC1 and silences developmental genes, while the second involves PCGF6-PRC1 and silences germ cell-specific and meiosis-specific genes (see the model in *Figure 7A*).

## The MAX-MGA heterodimer plays a role in recruiting PCGF6-PRC1 to target loci

By surveying DNA sequences of the PCGF6-PRC1 target genes, we find that they are enriched for the binding motif of the bHLHZ (basic helix-loop-helix, also containing a leucine zipper) transcription factor MAX (*Figure 4—figure supplement 1D*). Previous reports showed that MAX forms a heterodimer with MGA, another E-box binding protein (*Hurlin et al., 1999*). Interestingly, we identified MGA as a PCGF6 interacting factor by mass-spectrometry (*Figure 1A,B*), indicating that MAX and MGA may be directly involved in recruitment of PCGF6-PRC1 into target genes. Consistent with this hypothesis, both MAX and MGA were enriched at PCGF6-PRC1 target genes (*Figures 1G*, *4A and C*); and depletion of MAX or MGA by siRNA caused a reduction of PCGF6 binding to these loci (*Figure 4C*).

## The MAX/MGA heterodimer independently recruits PCGF6 and L3MBTL2

Previous studies show that the MAX/MGA heterodimer also plays a role in recruitment of the ncPRC1 component L3MBTL2 (*Suzuki et al., 2016*). Our results demonstrate that MAX/MGA independently associates with PCGF6 or L3MBTL2, as ablation of PCGF6 does not affect binding of L3MBTL2 (*Figures 3E* and *4D*). These findings indicate that MAX/MGA represses target genes through multiple mechanisms that may function in parallel. Consistent with this notion, ESCs deficient for MAX or MGA show a more drastic phenotype than the *Pcgf6*-KO ESCs (*Figure 6A,B,D*) (*Hishida et al., 2011*; *Washkowitz et al., 2015*).

Interestingly, L3MBTL2 associates with the H3K9me2 methyltransferases G9A and GLP (*Ogawa et al., 2002*; *Qin et al., 2012*). As G9A and GLP silence germ cell-specific genes in ESCs (*Maeda et al., 2013*), we speculate that L3MBTL2 contributes to transcriptional repression of these genes in association with G9A and GLP; while the same group of genes are silenced by a parallel pathway involving PCGF6 and RING1B (see the model in *Figure 7B*). The existence of several repressive epigenetic mechanisms that target germ cell-specific genes suggests the importance of preventing aberrant activation of such genes during pre- and peri-implantation embryonic development.

## Pleiotropic roles of PCGF6 in pre- and post-implantation embryonic development

Consistent with this idea, genetic ablation of *Pcgf6* leads to growth defects in both ESCs and epiLCs (*Figure 5C, D and E*), stem cells lines that approximate pre- and post-implantation embryonic developmental stages, respectively. Furthermore, constitutive deletion of *Pcgf6* causes impaired development of the placenta, in particular of the spongiotrophoblast layer and trophoblast giant cells (TGCs) (*Figure 6J and K*). Such aberrant phenotypes are also observed in the embryo proper,

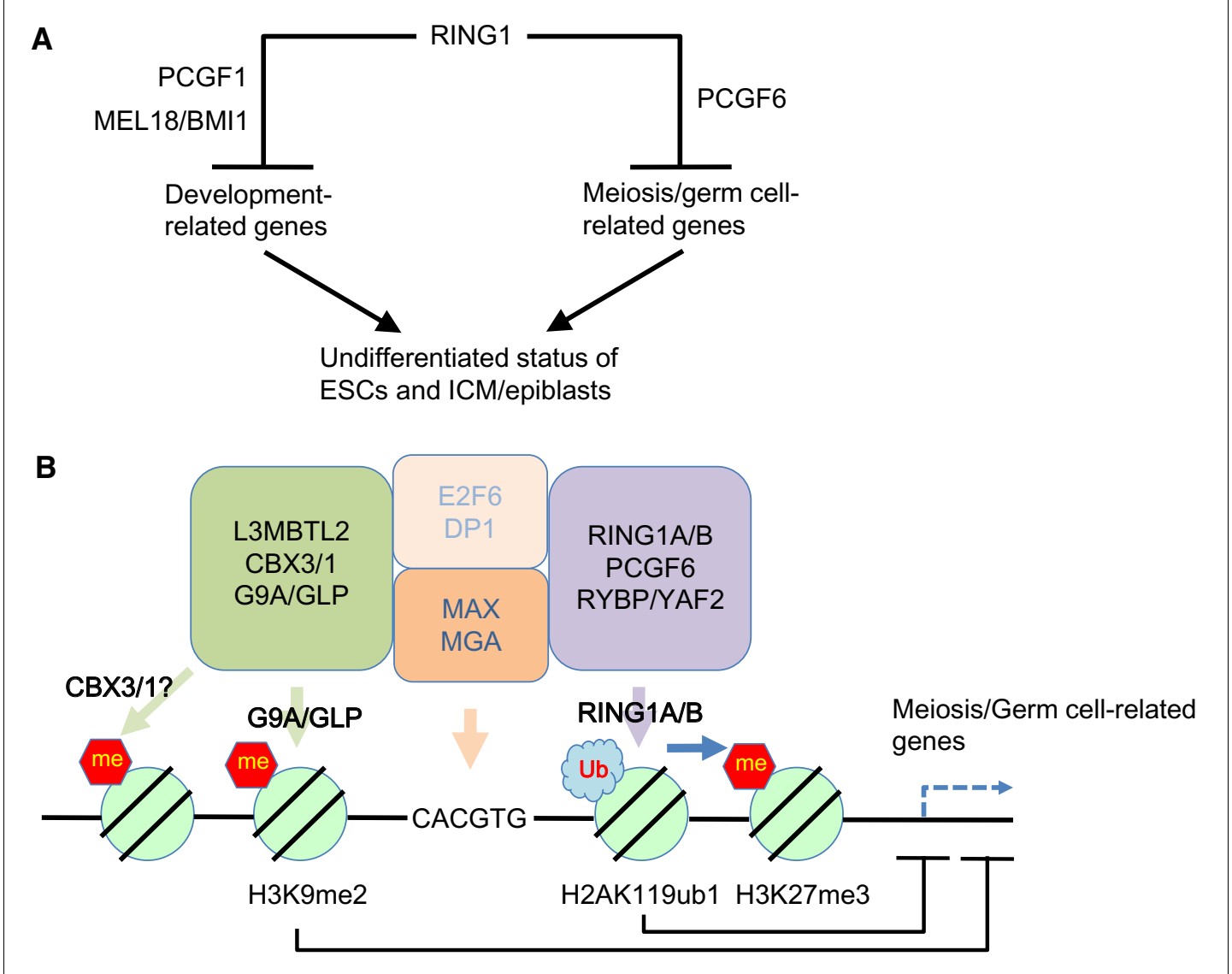

**Figure 7.** Our model for the role of PCGF6-PRC1. (**A**) Schematic representation for two major modes of gene silencing by PRC1. The first involves PCGF1-PRC1 (PCGF6) and cPRC1 (MEL18/BMI1), which silences developmental genes, and the second involves PCGF6-PRC1 (PCGF6), which silences germ cell-related and meiosis-related genes. (**B**) Schematic representation of PCGF6-PRC1-mediated regulation of germ cell-related and meiosis-related genes in ESCs and pre-/peri-implatation embryos.

including defects in anterior-posterior (A-P) axis specification and ectopic expression of developmental genes such as *Hoxb6* (*Figure 6G, H and I*).

## Concluding remarks

Taken together, we demonstrate a previously unappreciated function of the non-canonical PRC1 component PCGF6 for repression of germ cell-specific genes in ESCs. We further show that transcriptional silencing of such genes requires the function of RING1B and H2AK119ub1, but not PRC2 and H3K27me3. PCGF6-bound genes are enriched for short CGIs, frequently marked by methylated DNA, and are not bound by PCGF1-PRC1. Indeed, developmental genes, which are the main targets of PCGF1-PRC1, are not bound by PCGF6-PRC1, revealing distinct functional roles of these two ncPRC1 complexes. Consistent with this model, PCGF6-PRC1 is recruited to target sites by the MAX/MGA heterodimer, while PCGF1-PRC1 is recruited by the unmethylated CpG binding protein

KDM2B (*Farcas et al., 2012*). Our work thus shows a link between sequence specific DNA binding by the MAX/MGA heterodimer with PRC1-mediated transcriptional silencing of germ cell-specific genes by PCGF6, for regulation of pre- and post-implantation embryonic development.

## Materials and methods

### Construction of expression vectors

The full length mouse *Pcgf6*, mouse *Pcgf2*, mouse *Ring1B*, and mouse *Max* were PCR-amplified and inserted into a pCAG-IRES-Puro, a pCAG-IRES-BlaS, or a pCAG-IRES-HisD eukaryotic expression vector (Kindly gifts from Dr. Hitoshi Niwa in Kumamoto University) that was modified to express an N-terminal Flag and 2XStrep2 tag, an N-terminal 3xFlag tag, or an N-terminal HA tag. All PCR generated constructs were verified by sequencing. Mutations of the PCGF6 RING-finger domain (H155Y) and the MAX basic domain (L31V and E32D) were introduced into the wild type *Pcgf6* and *Max* constructs, respectively, using the PrimeSTAR Mutagenesis Basal Kit (Takara, Japan).

### Antibodies

The RING1B mouse monoclonal antibody has been described previously (*Atsuta et al., 2001*). A rabbit polyclonal antibody against the Flag-2XStrepII sequence was generated as described previously (*Farcas et al., 2012*). Commercially available antibodies were used to detect FLAG-tag (Sigma-Aldrich, St. Louis, MO,, F1804, RRID:AB_262044), HA-tag (Santa Cruz, Dallas, TX, sc-805), RYBP (Millipore, Billerica, MA, AB3637, RRID:AB_631618), L3MBTL2 (Active Motif, Carlsbad, CA, 39569, RRID:AB_2615062), MAX (Santa Cruz, sc-197X, RRID:AB_2281783), MGA (Bethyl, Montgomery, TX, A302-864A, RRID:AB_2615457), PCGF1 (Santa Cruz, E-8: sc-515371), PCGF2 (Santa Cruz, H-115: sc-10744, RRID:AB_2267885), PCGF6 (ORIGENE, Rockville, MD, TA324658), EZH2 (Cell Signaling, Danvers, MA, 4905, RRID:AB_2278249), SUZ12 (Cell Signaling, 3737, RRID:AB_2196850), SET1 (Bethyl, A300-289A, RRID:AB_263413), trimethylated Histone H3 lysine 27 (H3K27me3) (Millipore, 07–449, RRID:AB_310624), dimethylated histone H3 lysine 9 (H3K9me2) (MBL, Japan, MABI0317), H3K27ac (Cell Signaling, 8173, RRID:AB_2616015), monoubiquitinated histone H2A lysine 119 (H2AK119ub1) (Millipore, 05–678, RRID:AB_309899; Cell Signaling, 8240, RRID:AB_10891618), mouse IgM (Millipore, 12–488, RRID:AB_390193), histone H3 (Millipore, 07–690, RRID:AB_417398), and LAMIN B (Santa Cruz, sc-6216, RRID:AB_648156).

### Mice and ESC lines

Targeting vectors to knock out *Pcgf6* (*Figure 1—figure supplement 1A*) were introduced into R1 or M1 ESCs. Chimeras were generated by the morula aggregation method. Conditional mutants were mated with *Rosa26::CreERT2* (ERT2-Cre) transgenic mice purchased from Artemis Pharmaceuticals (Germany)(*Seibler et al., 2003*). ESCs were generated from respective blastocyst embryos as described previously (*Endoh et al., 2008*). Male ESCs were used in this study. 4-hydroxy tamoxifen (OHT) at a final concentration of 800 nM was added to culture medium to induce Cre recombinase activity and delete *Pcgf6*. *Cbx1/Cbx3* conditional KO ESCs (*Figure 3—figure supplement 1E*) were generated as described above. *Ring1A*$^{-/-}$; *Ring1B*$^{fl/fl}$; *Rosa26::CreERT2*$^{tg/+}$ ESCs (*Endoh et al., 2008*), *Pcgf2/4*-dKO ESCs (*Elderkin et al., 2007*), *Kdm2b*-KO ESCs (*Blackledge et al., 2014*), and *Max* conditional KO ESCs (*Hishida et al., 2011*) were described previously. For generation of ESC lines stably expressing exogenous cDNAs, 5 µg of the expression constructs were transfected into ESCs using electroporation and stable transfectants were selected using 1.0 µg/mL puromycin as described previously (*Endoh et al., 2012*). Prior to their use in ChIP experiments, ESCs were cultured for two passages under feeder-free conditions. All ESC lines used in this study were tested negative for mycoplasma before use. All animal experiments were carried out according to the in-house guidelines for the care and use of laboratory animals of the RIKEN Center for Integrative Medical Sciences, Yokohama, Japan [Approval number: Kei-27-001(7)]. All gene recombination experiments were carried out according to the in-house guidelines for the genetic recombination experiments of the RIKEN Center for Integrative Medical Sciences (Approval number: Sho-Y2015-018-10).

### Max/Mga knockdown

The siRNAs for *Max* and *Mga* were selected according to a previous report (*Maeda et al., 2013*) and purchased from Qiagen (Germany). ES cells were transfected with individual siRNAs using Lipofectamine RNAiMAX reagent (Thermo Fisher Scientific, Waltham, MA) according to the manufacturer's instructions.

### PCGF6 protein complex purification

To purify PCGF6 and associated proteins, a mouse ESC line stably expressing Flag-2XStrepII-tagged PCGF6 was generated. Nuclear extracts were isolated from this cell line, the PCGF6 complex was affinity purified, and the co-purified proteins were subject to mass spectrometry as described previously (*Farcas et al., 2012*).

### Immunoprecipitation (IP) analysis

Cells expressing each of the tagged constructs were suspended in IP buffer [10 mM Tris-HCl (pH8.0), 1 mM EDTA, 140 mM NaCl, 0.4% NP-40, and 0.5 mM PMSF] and sonicated for several seconds. After centrifugation, the supernatant was collected, and then incubated with anti-FLAG antibody (Sigma-Aldrich; M2) for 120 min at 4°C. The immune complexes were captured by incubation with protein A/G magnetic Dynabeads (Invitrogen, Carlsbad, CA) for 60 min at 4°C. The bead-bound proteins were washed with IP buffer, eluted in SDS sample buffer under reducing condition, separated on SDS-PAGE gels, and subjected to western blot analysis. Tandem mass spectrometry (LC–MS/MS) was performed as described previously (*Farcas et al., 2012*). Materials eluted with desthiobiotin were collected and precipitated using chloroform/methanol and re-suspended materials were subjected to in-solution tryptic digestion followed by nano-liquid chromatography-tandem mass spectrometry (nLC-MS/MS) analysis using a nano-Acquity UPLC (Waters, Milford, MA) coupled to an Orbitrap Velos/Elite mass spectrometer (Thermo Fisher Scientific). MS/MS spectra were searched against the UniProt SwissProt Mouse database (16,683 sequences) in Mascot v2.3.01. Protein assignment was based on at least two identified peptides. Mascot scores and peptide coverage are indicated for each protein (*Figure 1B*).

### Gene expression analysis

Total RNA was extracted using the RNeasy Mini kit (Qiagen) with the RNase-Free DNase Set (Qiagen) according to the manufacturer's instructions. Genomic DNA-free RNA samples were further purified using the RNeasy kit RNA cleanup protocol. For subsequent RT-PCR analysis, cDNA was synthesized with the SuperScript III First-Strand Synthesis System (Thermo Fisher Scientific). Quantitative real-time PCR was performed by using the Brilliant III SYBR Green QPCR Master Mix (AgilentLaboratories, Santa Clara, CA), using *Gapdh* as housekeeping gene. For RNA-seq studies, RNA integrity was assessed on a BioAnalyzer; all samples had a RNA Integrity Number (RIN) $\geq$9.5 (Agilent Laboratories). Sequencing libraries were generated according to Illumina's instructions and sequenced on the Illumina HiSeq platform as described previously (*Sharif et al., 2016*).

### ChIP-qPCR analysis

Chromatin immunoprecipitation (ChIP) was performed as previously described (*Endoh et al., 2012*), with minor modifications. Sonication was performed using a Covaris focused-ultrasonicator (Covaris, Woburn, MA) to produce fragments of approximately 0.5–1 kb. Immunoprecipitation was performed overnight at 4°C with approximately 2–5 µg of antibody and chromatin corresponding to $5 \times 10^6$ cells. Antibody bound proteins were isolated on protein A/G magnetic Dynabeads (Invitrogen), washed extensively, eluted, cross-links were reversed, and then the samples were sequentially treated with RNase and proteinase K before being purified using phenol-chloroform extraction. Real-time qPCR was performed using the Brilliant III SYBR Green QPCR Master Mix (Agilent). For H2AK119u1-ChIP using E6C5 (Millipore #05–678), pre-cleared chromatin from $2–3 \times 10^6$ cells was incubated with 40 ul of E6C5 antibody (Millipore #05–678) (overnight, 4°C), and 30 ul of original Protein A dynabead slurry (Invitrogen) was incubated with 3 ug (=3 ul) of rabbit anti-mouse IgM antibody (Millipore #12–488) (overnight, 4°C). Next day, the chromatin-1st antibody complexes were immunoprecipitated with second antibody - preconjugated protein A dynabeads, and subjected to the conventional ChIP experiments as described above.

## ChIP-seq analysis

ChIP-seq libraries were prepared according to Illumina's instructions accompanying the NEBNext ChIP-Seq Library set (NEB, Ipswich, MA, #E6200) and quantified by the KAPA Library Quantification Kit (KAPA, Wilmington, MA), and their sizes were confirmed by Bioanalyzer (Agilent). Libraries were sequenced using Illumina (San Diego, CA) HiSeq system as described previously (*Isono et al., 2013*). The peak calling of ChIP-seq data was performed with MACS2 program. Published ChIP-seq data for CBX7, MAX, and MYC were obtained from NCBI GEO (accession numbers GSM1041373, GSM1171650, GSM1171648, respectively). Published ChIP-seq data for RING1B in wild-type and *Kdm2b*-KO ESCs were also obtained from NCBI GEO (Series GSE55698).

## Motif analysis

Binding motifs of PCGF6, RING1B, CBX7, MAX and MYC were detected using a computer program DREME included in the motif analysis suite MEME version 4.9.1 (http://meme-suite.org/). Briefly, 500 bp sequences around the centers of ChIP-seq peaks were collected from the mouse genome and enriched motifs were calculated using the above program with default parameters. Graphical representation of the sequence logo, where letter height was determined by Shannon entropy, was generated using our in-house program.

## Generation of the de novo targeting system

E14 mouse ES cells containing a single copy of TetO BAC were generated as described previously (*Blackledge et al., 2014*). Mouse Pcgf6 or Max coding sequence was inserted into pCAGFS2TetR to generate a mammalian expression plasmid for N-terminal FLAG STREPx2 (FS2) tagged PCGF6 or MAX, respectively. Either of these plasmids was transfected into the TetO-containing ESCs and stable clones expressing TetR-PCGF6 or TetR-MAX fusion protein were detected in ChIP experiments using an FS2-specific antibody as previously described (*Blackledge et al., 2014*).

## Induction of EpiLCs and PGCLCs

PGCLCs were induced from *Pcgf6*$^{fl/fl}$ ES cells through an epiblast-like state as described previously (*Hayashi et al., 2011*; *Hayashi and Saitou, 2013*). Briefly, the ESCs were first cultured in 2i (MEK inhibitor PD0325901 and GSK-3 inhibitor CHIR99021)-containing media without a feeder layer for 3 days (from day −5 to day −2 in *Figure 5C*) and then subjected to EpiLC induction culture by using media containing Activin A and bFGF for 2 days (from day −2 to day 0). Induced EpiLCs were further differentiated into the germ cell-linage by culturing them with BMPs, SCF, LIF, EGF and others for 6 days (from day 0 to day 6). The impact of PCGF6 loss at different stages was tested by adding OHT at day −5, day −2, or day +2 (*Figure 5C*).

## Histological and skeletal analysis of *Pcgf6*-KO mice

Skeletal preparations were made from newborn mice and cleared skeletons were analyzed under a stereomicroscope as described previously (*Akasaka et al., 1996*). RNA in situ hybridization was performed as described previously (*Akasaka et al., 1996*).

## Data access

Sequencing and microarray data can be accessed via the geo accession GSE84480 (http://www.ncbi.nlm.nih.gov/geo/query/acc.cgi?token=wdmlougulbwbtgr&acc=GSE84480) and GSE87484 (https://www.ncbi.nlm.nih.gov/geo/query/acc.cgi?token=sbebmckgtlyhlat&acc=GSE87484).

## Acknowledgements

We thank Dr. Natsumi Shimizu in RIKEN CDB for providing technical advice for mouse early embryos. This work was supported by Grant-in-Aid for Scientific Research on Innovative Areas (#26112516 to ME), Grant-in-Aid for Young Scientist (B) (# 25871129 to ME) and Grant-in-Aid for Scientific Research (C) (#16K07372 to ME) from Ministry of Education, Culture, Sports, Science and Technology (MEXT) of Japan. HK is a recipient of the Strategic Basic Research Programs, CREST grant.

## Additional information

### Funding

| Funder | Grant reference number | Author |
|---|---|---|
| Ministry of Education, Culture, Sports, Science, and Technology | Grant-in-Aid for Young Scientist (B) (#25871129) | Mitsuhiro Endoh |
| Ministry of Education, Culture, Sports, Science, and Technology | Grant-in-Aid for Scientific Research (C) (#16K07372) | Mitsuhiro Endoh |
| Ministry of Education, Culture, Sports, Science and Technology | Grant-in-Aid for Scientific Research on Innovative Areas (#26112516) | Mitsuhiro Endoh |
| Ministry of Education, Culture, Sports, Science and Technology | Grant-in-Aid for Scientific Research on Innovative Areas (#24116711) | Mitsuhiro Endoh |
| RIKEN | | Haruhiko Koseki |
| Ministry of Education, Culture, Sports, Science, and Technology | | Haruhiko Koseki |
| Japan Science and Technology Agency | Strategic Basic Research Programs | Haruhiko Koseki |

The funders had no role in study design, data collection and interpretation, or the decision to submit the work for publication.

### Author contributions

ME, Conceptualization, Resources, Data curation, Funding acquisition, Validation, Investigation, Writing—original draft, Writing—review and editing; TAE, Data curation, Validation, Investigation, Visualization, Writing—original draft, Writing—review and editing; JShi, Resources, Data curation, Investigation, Writing—review and editing; KH, Data curation, Investigation, Writing—original draft; AF, K-WM, TE, TI, BMK, Resources, Investigation; SI, NO, Investigation; JSha, Data curation, Investigation, Writing—original draft, Writing—review and editing; MN, AO, Resources; OM, Data curation, Investigation; TS, Supervision, Writing—review and editing; OO, Supervision; RK, Resources, Supervision, Investigation; HK, Conceptualization, Resources, Data curation, Supervision, Funding acquisition, Investigation, Writing—original draft, Writing—review and editing

### Author ORCIDs

Mitsuhiro Endoh, http://orcid.org/0000-0003-0715-1701
Akihiko Okuda, http://orcid.org/0000-0002-5298-5564
Robert Klose, http://orcid.org/0000-0002-8726-7888
Haruhiko Koseki, http://orcid.org/0000-0001-8424-5854

### Ethics

Animal experimentation: All animal experiments were carried out according to the in-house guidelines for the care and use of laboratory animals of the RIKEN Center for Integrative Medical Sciences, Yokohama, Japan [Approval number: Kei-27-001(7)].

## Additional files

### Supplementary files

• Supplementary file 1. The sequences of primers used in quantitative ChIP-PCR and RT-PCR.

### Major datasets

The following datasets were generated:

| Author(s) | Year | Dataset title | Dataset URL | Database, license, and accessibility information |
|---|---|---|---|---|
| Endoh M, Koseki H, Sharif J, Endo TA | 2017 | PCGF6-PRC1 suppresses premature differentiation of embryonic stem cells by silencing germ cell-related genes [RNA-Seq] | https://www.ncbi.nlm.nih.gov/geo/query/acc.cgi?acc=GSE84480 | Publicly available at the NCBI Gene Expression Omnibus (accession no: GSE84480) |
| Endoh M, Koseki H, Sharif J, Endo TA | 2017 | PCGF6-PRC1 suppresses premature differentiation of embryonic stem cells by silencing germ cell-related genes [ChIP-Seq] | https://www.ncbi.nlm.nih.gov/geo/query/acc.cgi?acc=GSE87484 | Publicly available at the NCBI Gene Expression Omnibus (accession no: GSE87484) |

The following previously published datasets were used:

| Author(s) | Year | Dataset title | Dataset URL | Database, license, and accessibility information |
|---|---|---|---|---|
| Croce LD | 2013 | Cbx7_ChIPSeq | https://www.ncbi.nlm.nih.gov/geo/query/acc.cgi?acc=GSM1041373 | Publicly available at the NCBI Gene Expression Omnibus (accession no: GSM1041373) |
| Neri F | 2014 | Max_ChIPSeq | http://www.ncbi.nlm.nih.gov/geo/query/acc.cgi?acc=GSM1171650 | Publicly available at the NCBI Gene Expression Omnibus (accession no: GSM1171650) |
| Neri F | 2014 | BioMyc_ChIPSeq | https://www.ncbi.nlm.nih.gov/geo/query/acc.cgi?acc=GSM1171648 | Publicly available at the NCBI Gene Expression Omnibus (accession no: GSM1171648) |
| Blackledge NP | 2014 | KDM2Bfl/fl_RING1B_ChIPSeq | https://www.ncbi.nlm.nih.gov/geo/query/acc.cgi?acc=GSE55698 | Publicly available at the NCBI Gene Expression Omnibus (accession no: GSE55698) |

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
