## [Decision Letter]

Thank you for submitting your article "PCGF6-PRC1 suppresses premature differentiation of embryonic stem cells by silencing germ cell-related genes" for consideration by *eLife*. Your article has been reviewed by three peer reviewers, and the evaluation has been overseen by Robb Krumlauf as a Reviewing Editor and Kevin Struhl as the Senior Editor. The following individual involved in review of your submission has agreed to reveal his identity: Giacomo Cavalli (Reviewer #3).

The reviewers have discussed the reviews with one another and the Reviewing Editor has drafted this decision to help you prepare a revised submission.

This study presents an interesting investigation into the role of the PRC1 cofactor PCGF6 in PRC1-mediated gene silencing. Following up on the recent characterization of non-canonical PRC1 sub-complexes the authors investigate the function of one such sub-complex containing the PCGF6 protein. They present data suggesting that PCGF6 is involved in the recruitment of non-canonical PRC1 to a subset of genes that encode germ cell regulators. Mechanistically, they propose that recruitment of PCGF6-PRC1 is mediated by MAX/MGA transcription factors.

Although the subject of the study is important in the field of Polycomb biology, the data presented lack clear evidence for a role of MAX in recruitment. It remains mostly correlative and does not provide direct evidence that MAX/MGA transcription factors recruit PCGF6-PRC1. There are a number of other concerns, controls and clarifications that need to be addressed in a revision before the study can be considered for publication.

1) A major concern of the work is that the evidence for a role of MAX in specifying PCGF6-PRC1 recruitment is overstated. Motifs enriched within PCGF6 binding sites are not exclusive to MAX. The effects of MAX knockdown on gene expression and observations based on the DNA-binding mutant of MAX do not rule out an indirect mode of action. There is no demonstration direct MAX-mediated recruitment of PCGF6-PRC1 with the enforced tethering system used in the study. The mass spectrometry data uncover an interaction with MGA, but not MAX. Hence, one of the major conclusions of the study is not adequately demonstrated by the data presented in the manuscript. The authors must demonstrate direct MAX-mediated recruitment of PCGF6-PRC1 with the enforced tethering system or alternative approach. If a Tet-MAX can recruit the PCGF6 ncPRC1 complex this would support their conclusions.

2) In Figure 1, it would help if a control Flag-Strep ChIP seq (i.e. of an FS2-empty vector cell line) is included alongside the FS2-Pcgf6 ChIP seq to understand the background signal in the experiment. This would ensure that all of the Pcgf6 signal shown is real. The reason for this is that sequencing of input chromatin alone often shows weak enrichment at transcription start sites relative to other regions of chromatin. Therefore, this requested control ChIP-seq would eliminate any doubt about the signal shown for FS2 tagged Pcgf6.

3) A suggestion to improve the paper would be to move the Figure 6—figure supplement 1 to the main figures as this data unequivocally shows the biological importance of Pcgf6. The developmental defects identified in the absence of Pcgf6 in vivo are much starker than the in vitro differentiation defects currently featured in Figure 6.

4) In the subsection “PCGF6 forms complexes with PRC1 components”, the authors purify the PCGF6-PRC1 complex and report copurification of CBX1 and CBX3 (HP1) proteins. This is an intriguing result that could link this complex to H3K9me3 and heterochromatin. Later, they confirm the presence of L3MBTL2 at GF6-PRC1 target genes. In the second paragraph of the subsection “PCGF6 recruits RING1B to target genes and facilitates downstream H2K119ub1 deposition” and Figure 3 the authors show ChIP-seq of RING1B in WT and Pcgf6-KO ESCs. In light of the presence of HP1 proteins (as well as G9a and GLP, at least following Ogawa et al. 2002) in the PCGF6-PRC1 complex, it would be really important to perform ChIP-seq of H3K9me2 in the same condition. Likewise, KD of CBX1 and CBX3 followed by RT-PCR would help understand whether K9me2-HP1 repressive chromatin does contribute to silencing of PCGF6-PRC1 target genes.

---

## [Author Response]

*[…] Although the subject of the study is important in the field of Polycomb biology, the data presented lack clear evidence for a role of MAX in recruitment. It remains mostly correlative and does not provide direct evidence that MAX/MGA transcription factors recruit PCGF6-PRC1. There are a number of other concerns, controls and clarifications that need to be addressed in a revision before the study can be considered for publication.*

*1) A major concern of the work is that the evidence for a role of MAX in specifying PCGF6-PRC1 recruitment is overstated. Motifs enriched within PCGF6 binding sites are not exclusive to MAX. The effects of MAX knockdown on gene expression and observations based on the DNA-binding mutant of MAX do not rule out an indirect mode of action. There is no demonstration direct MAX-mediated recruitment of PCGF6-PRC1 with the enforced tethering system used in the study. The mass spectrometry data uncover an interaction with MGA, but not MAX. Hence, one of the major conclusions of the study is not adequately demonstrated by the data presented in the manuscript. The authors must demonstrate direct MAX-mediated recruitment of PCGF6-PRC1 with the enforced tethering system or alternative approach. If a Tet-MAX can recruit the PCGF6 ncPRC1 complex this would support their conclusions.*

We thank you for these critical comments and suggestions. Although we failed to detect MAX in our mass spec analysis, we found a considerable association of MAX with FLAG-PCGF6 by IP-immunoblotting with several negative controls as shown in the revised Figure 1. In addition, we further showed co-existence of MAX with PCGF6 mainly at PCGF6-bound genes irrespective of RING1B binding as shown in Figure 1. We further confirmed the enforced tethering of PCGF6 efficiently brought MAX as shown in the revised Figure 3. We have revised the text accordingly.

Upon the request, we further examined whether the enforced tethering of MAX could recruit PCGF6-PRC1. We expressed MAX/TETR and ectopically tethered this fusion protein to a pre-integrated TetO array in ESCs (Figure 4—figure supplement 2). ChIP-qPCR analysis showed binding of the MAX/TETR fusion protein at the TetO array, which accompanies mild enrichments of HA-tagged PCGF6, H2AK119ub1 and H3K27me3 (Figure 4—figure supplement 2) while barely RING1B (data not shown). These data again support MAX-dependent recruitment of PCGF6-PRC1 though its impact is limited in this experimental setup. We guess this limited effect of MAX/TETR to recruit PCGF6 could be due to a potential dimerization of MAX with other partners than MGA in this experimental setup. We revised the main text (subsection “Interactions between MAX and PCGF6 play a role in ESC maintenance”, last paragraph) accordingly.

*2) In Figure 1, it would help if a control Flag-Strep ChIP seq (i.e. of an FS2-empty vector cell line) is included alongside the FS2-Pcgf6 ChIP seq to understand the background signal in the experiment. This would ensure that all of the Pcgf6 signal shown is real. The reason for this is that sequencing of input chromatin alone often shows weak enrichment at transcription start sites relative to other regions of chromatin. Therefore, this requested control ChIP-seq would eliminate any doubt about the signal shown for FS2 tagged Pcgf6.*

Thank you for this suggestion. We included a control Flag ChIP-seq data (obtained from a Flag-empty vector cell line) alongside the Flag-PCGF6 data in Figure 1 (shown as NC) to ensure the Flag-PCGF6 signal shown is real.

*3) A suggestion to improve the paper would be to move the Figure 6—figure supplement 1 to the main figures as this data unequivocally shows the biological importance of Pcgf6. The developmental defects identified in the absence of Pcgf6* in vivo *are much starker than the* in vitro *differentiation defects currently featured in Figure 6.*

We appreciate your suggestion and moved these figures to Figure 6 in the revised version.

*4) In the subsection “PCGF6 forms complexes with PRC1 components”, the authors purify the PCGF6-PRC1 complex and report copurification of CBX1 and CBX3 (HP1) proteins. This is an intriguing result that could link this complex to H3K9me3 and heterochromatin. Later, they confirm the presence of L3MBTL2 at GF6-PRC1 target genes. In the second paragraph of the subsection “PCGF6 recruits RING1B to target genes and facilitates downstream H2K119ub1 deposition” and Figure 3 the authors show ChIP-seq of RING1B in WT and Pcgf6-KO ESCs. In light of the presence of HP1 proteins (as well as G9a and GLP, at least following Ogawa et al. 2002) in the PCGF6-PRC1 complex, it would be really important to perform ChIP-seq of H3K9me2 in the same condition. Likewise, KD of CBX1 and CBX3 followed by RT-PCR would help understand whether K9me2-HP1 repressive chromatin does contribute to silencing of PCGF6-PRC1 target genes.*

Thank you for your important suggestion. We performed ChIP-seq for H3K9me2 and the summary of data is included in Figure 3—figure supplement 1. Unlike L3mbtl2-KO ESCs (Qin et al., 2012), we did not observe considerable changes in H3K9me2 levels in Pcgf6-KO. Consistent with this we also showed L3MBTL2 bound to the targets in Pcgf6-KO. We suggest L3MBTL2/G9A axis acts in parallel with PCGF6/RING1 pathway within the same complexes. Indeed, we also found L3MBTL2 bound to the targets in MAX-dependent manner. MAX/MGA heterodimer likely recruit PCGF6 and L3MBTL2 mutually independently as summarized in Figure 7.

We also performed RNA-seq analysis of Cbx1/3-double knockout (dKO) ESCs to investigate the role of CBX1/3 for the regulation of PCGF6-PRC1 targets (Figure 3—figure supplement 1). We did not observe a significant change in the expression levels of PCGF6+CBX7-RING1B+ in Cbx1/3-dKO ESCs, suggesting CBX1/3 have, if any, a marginal role for the repression of PCGF6-PRC1 targets (Figure 3—figure supplement 1).

We revised the main text (subsection “PCGF6 recruits RING1B to target genes and facilitates downstream H2AK119ub1 and H3K27me3 deposition”, last paragraph) accordingly.